Optimization of a gas chromatographic unit for measuring BVOCs in ambient air

Kenneth Mermet[1,2,3], Stéphane. Sauvage[1], Sébastien Dusanter[1], Thérèse Salameh[1], Thierry Léonardis[1], Pierre-M. Flaud[2,3], Émilie Perraudin[2,3], Éric Villenave[2,3], Nadine Locoge[1]

[1] IMT Lille Douai, Univ. Lille, SAGE - Département Sciences de l'Atmosphère et Génie de l'Environnement, 59000 Lille, France
[2] Univ. Bordeaux, EPOC, UMR 5805, F-33405 Talence Cedex, France
[3] CNRS, EPOC, UMR 5805, F-33405 Talence Cedex, France

*Correspondence to*: Stéphane Sauvage (stephane.sauvage@imt-lille-douai.fr) or Nadine Locoge (nadine.locoge@imt-lille-douai.fr)

**Abstract.** A new online gas chromatographic method dedicated to Biogenic Volatile Organic Compounds (BVOC) analysis was developed for the measurement of a 20 BVOC gaseous mixture (isoprene, β-pinene, α-pinene, limonene, trans-β-ocimene, myrcene, sabinene, $\Delta^3$-carene, camphene, 1,8 cineole, terpinolene, linalool, α-phellandrene, nopinone, citral, α-terpinene, β-caryophhyllene, p-cymene, γ-terpinene and 2-carene) at a time resolution of 90 minutes. The optimized method includes an online Peltier-cooled thermodesorption system sample trap made of Carbopack B coupled to a gas chromatographic system equipped with a 60 m, 0.25 mm i.d. BPX5 column. Eluent was analysed using flame detection (FID). Potassium iodide was identified as the best ozone scrubber for the 20 BVOC mixture. In order to obtain an accurate quantification of BVOC concentrations, the development of a reliable standard mixture was also required. Quantification of BVOCs was reported with a detection limit ranging from 4 ppt for α-pinene to 19 ppt for sabinene. The main source of uncertainty was the calibration step, stressing the need of certified gaseous standards for a wider panel of BVOCs. This new method was applied for the first time to measure BVOCs in a pine forest during the LANDEX-episode-1 field campaign (summer 2017). All targeted BVOCs were detected at least once along the campaign. The two major monoterpenes observed were β-pinene and α-pinene, representing on average 60% of the measured terpenoid concentration, while isoprene represented only 17%. Uncertainties determined were always below 13% for the six major terpenes.

**Copyright statement**

**1 Introduction**

Emissions of volatile organic compounds (VOCs) can impact both (i) the atmospheric oxidation capacity (Houweling et al., 1998; Lelieveld et al., 2008; Taraborrelli et al., 2012), due to the reactivity of VOCs with atmospheric oxidants such as ozone ($O_3$), hydroxyl (OH) and nitrate ($NO_3$) radicals (Atkinson and Arey, 2003), and (ii) the earth's radiative balance

(Gauss et al., 2006; IPCC, 2013; Hoffmann et al., 1997; Kazil et al., 2010) through the formation of ozone and secondary organic aerosols (SOAs).

Biogenic VOCs (BVOCs) represent the largest fraction of non-methane VOCs emitted in the troposphere, contributing to 75-90% of the total global emissions (Guenther et al., 1995; Lamarque et al., 2010; Sindelarova et al., 2014). Global BVOC emissions are composed at 87% of terpenes (Messina et al., 2016) covering a wide range of volatility, including isoprene ($C_5H_8$), monoterpenes ($C_{10}H_{16}$), sesquiterpenes ($C_{15}H_{24}$) and some oxygenated terpenoids (Kesselmeier and Staudt, 1999). Each of these groups of compounds exhibits a large number of structural isomers, with a large density of reactivity.

The impact of BVOC emissions on the carbon cycle and the atmospheric oxidant budget at both local and global scales is currently not well understood. Indeed, reported measurements of total OH reactivity performed in ambient air highlighted some gaps in our knowledge about OH sinks, especially in forested regions, where the measured OH reactivity is frequently higher than that calculated from concomitant VOC observations (Carslaw et al., 2001; Di Carlo et al., 2004; Dusanter and Stevens, 2017; Edwards et al., 2013; Griffith et al., 2013; Hansen et al., 2014; Hens et al., 2014; Stavrakou et al., 2010; Tan et al., 2001; Wolfe et al., 2014; Zannoni et al., 2017). This difference reveals the presence of unmeasured OH sinks within the forest boundary layer, which may either be attributed to unidentified primary BVOC emissions (Di Carlo et al., 2004; Sinha et al., 2010), oxidation products of BVOCs (Edwards et al., 2013; Hansen et al., 2014; Lou et al., 2010; Mao et al., 2012; Zannoni et al., 2017) or both (Nölscher et al., 2012). Also, in those studies, a potential underestimation of concentration, high uncertainties on BVOC concentrations, and ozone reactivity (when no scrubber was used) could explain this missing reactivity. Speciated measurements of these compounds are therefore important to improve our understanding of the atmospheric composition and reactivity.

During intensive field campaigns, isoprene and terpene concentrations are usually investigated using proton transfer reaction mass spectrometry, a very efficient and fast technique (with a time resolution better than one second) but only providing information about the sum of monoterpenes (Bouvier-Brown et al., 2009; de Gouw and Warneke, 2007; Park et al., 2014; Zhou et al., 2017, Kammer et al., 2018) and sesquiterpenes (Bouvier-Brown et al., 2009, 2007; Kim et al., 2009; Park et al., 2014; Zhou et al., 2017), as it cannot distinguish individual structural isomers. This type of instrument has recently been coupled to a fast gas-chromatography (GC) to separate monoterpenes (Materić et al., 2015), the feasibility for ambient measurements has yet to be demonstrated.

Detailed information regarding chemical composition may be obtained by conventional gas chromatographic methods (GC-FID or GC-MS), which can quantify individual terpene isomers (Jones et al. 2014; Hopkins et al., 2011; Pankow et al. 2012). Ambient measurements taking advantage of gas chromatographic techniques usually report isoprene, α-pinene, β-pinene and limonene (Apel et al., 1999; Bouvier-Brown et al., 2009; Greenberg et al., 2004, 1999; Hopkins et al., 2011; Mallet et al., 2016; Misztal et al., 2010; Saxton et al., 2007) as the major BVOCs. Only a few GC instruments have been optimized to provide a larger speciation of monoterpenes (Pankow et al., 2012; Jones et al., 2014; Hakola et al., 2006, 2017) and even

fewer can provide a large speciation of both monoterpenes and oxygenated monoterpenes (Jones et al., 2014; Pankow et al., 2012) (Table 1).

**Table 1 : Comparison of operated conditions (sampling method, detector and column used) and limit of detection (DL) (number of species follow)**

| | This study | Hakola et al. (2017) | Jones et al. (2014) | Pankow et al. (2012) | Hopkins et al. (2011) | Hakola et al. (2006) | Greenberg et al. (2004) |
|---|---|---|---|---|---|---|---|
| On-line/not | on-line | on-line | on-line | not on-line | on-line | not on-line | not online |
| Collection | ATD, 12 liter | ATD, 1 liter | ATD, 0.75 liter | ATD, 5 liter | ATD, 1 liter | ATD, 3 liter | ATD, 6 liter |
| composition trap | Carbopack B | Tenax TA Carbopack B | Tenax | Tenax-TA and Carbotrap B or Tenax GR and Carbograph | Carboxen 1000 and Carbotrap B (90 mg) | Tenax-TA carbopack-B | glass beads (80 mg), Carbotrap B (170 mg), Carbosieve III (350 mg) |
| Detection | GC/FID | TDGC-MS | GC/FID | GCxGC ToFMS | GCxGC FID | GC/MS | GC/MS |
| Column dimension | BPX-5 60 m, 0.25 mm i.d.,1µm | DB-1 60 m, 0.25 mm i.d., 0.25 µm | MXT-5 15m, 0,.25mm i.d., 0.25µm | DB-VRX, Stabilwax 45m,0.25mm id, .,4µm; 1.5m, 0.25mm id, 0.25µm | PLOT, LOWOX 50m,0.53mm id; 2x10m, 0.3mm id | HP-1 60 m, 0.25 mm i.d. | DB-1 30 m, 0.32 mm i.d., 1µm |
| LoD definition | Signal/noise (S/N) = 3 | not stated | not stated | S/N = 10 | not stated | not stated | not stated |
| compound | pptv | pptv | pptv | pptv | pptv | pptv | pptv |
| Isoprene | 12 | | | 4 | 1 | 11.4 | 1 |
| monoterpenes | 4-19 (14) | <1(8) | 4-5 (12) | 0.7-2.1(9) | 3 (5) | 5.2-10.7 (7) | 1(4) |
| Oxygenated monterpenes | 4-11(4) | | 4 (4) | 13-19 (6) | | 13.2 (1) | |
| Oxidation product | 7 (1) | <1(1) | 5 (2) | 2.1(1) | | | |
| sesquiterpenes | 9 (1) | <1(6) | | 0.9-1.4 (4) | | 9.4 (1) | |

Due to their high reactivity, monoterpenes and sesquiterpenes are delicate to quantify. Amongst potential artefacts, ozonolysis of monoterpenes and sesquiterpenes within the sampling line may occur and the use of ozone scrubbers is recommended (Koppmann, 2007). Some scrubbers have already been used for BVOC measurement, such as heated stainless steel tube (Hellen et al., 2012), copper tubes coated by potassium iodide (KI) (Saxton et al., 2007) and thiosulfate scrubbers (Bouvier-Brown et al., 2007; Jones et al., 2014; Plass-Dülmer et al., 2002).

Conservation of air samples (in pressured gas canisters or adsorbed on cartridges) for offline analysis of monoterpenes and sesquiterpenes needs to be carefully considered (Apel et al., 1999; Greenberg et al., 2004; Misztal et al., 2010; Pankow et al., 2012). Due to their reactivity or poor stability, certified gas standard mixtures containing multiple terpenes are not readily available, and hence calibration of these gases is generally less straightforward compared to that of other NMHCs. Rhoderick and Lin (2013) demonstrated that 20 L aluminium canisters with proprietary internal coatings are capable of containing gaseous monoterpenes in nitrogen without significant degradation for periods of >250 days; however further investigations are necessary to ensure consistency between canisters, and to test whether this level of stability may be achieved for gaseous mixtures containing both α- and β-pinene. With the exception of isoprene, α- and β-pinene, $\Delta^3$-carene, myrcene, limonene and eucalyptol, certified gas cylinder containing other terpenes are not available. To quantify other terpenes, two approaches may be used. As flame ionization detector response is a very stable and linear method, over several orders of magnitude, Structure Activities Relationships (SAR) and Effective Carbon Number (ECN) can be used with a reference compound (like toluene) (Hopkins et al., 2011). Otherwise, a liquid solution of pure (≥ 95%) compounds can be

vaporised in a gaseous flow or deposited on a clean cartridge (Bouvier-Brown et al., 2009, 2007; Hakola et al., 2017, 2006; Pankow et al., 2012).

This study reports the optimization of a fully automated online GC-FID instrument for ambient measurements of a series of twenty primary BVOCs and some of their oxidation products, including isoprene, 17 monoterpenes (w/ 3 oxygenated species), nopinone and β-caryophyllene. The choice of sampling (ozone scrubber, sampling flow rate, sampling duration) and analytical (thermodesorption temperature, GC column) parameters is discussed below. A calibration method is also proposed for BVOCs that are not available in commercial standards. The first deployment of this instrument in a pine forest during the LANDEX episode 1 campaign in summer 2017 is presented together with an evaluation of analytical performances for every targeted species.

## 2 Materials and methods

The online TD-GC-FID system used in this study was composed of a 6890N chromatograph (Agilent) and a Unity 1 air sampler (Markes). Ambient air was sampled through an ozone scrubber and passed into a trap held at 20°C and filled with Carbopack B (Supelco) for 60 minutes. After this pre-concentration step, VOCs were thermodesorbed and injected in a BPX-5 column (SGE Analytical Science) where the compounds were separated over 67 minutes. The measurement method was optimized to provide a time resolution of 90 minutes. Details about the material used and the optimization of operating conditions are given below.

### 2.1 Targeted species and gas standards

A list of 20 targeted species was selected for various reasons: some are observed to be the most abundant species at the global scale (isoprene, β-pinene, α-pinene, limonene, trans-β-ocimene, myrcene, sabinene, $\Delta^3$-carene and camphene) (Guenther et al., 1995; Sindelarova et al., 2014), others are emitted by pine trees (1,8 cineole) (Simon et al., 1994), some are present in pine needles (terpinolene, linalool, α-phellandrene) (Ait Mimoune et al., 2013; Arrabal et al., 2012; Blanch et al., 2012; Kleinhentz et al., 1999; Ormeño et al., 2009; Simon et al., 1994), others are oxidation products, commercially available in pure solution (≥ 95%) of monoterpenes (nopinone, citral), and some are highly reactive (α-terpinene, β-caryophyllene) (Atkinson et al, 2006) or usually monitored along with other BVOCs (p-cymene, γ-terpinene, 2-carene) (Hakola et al., 2017; Jones et al., 2014; Pankow et al., 2012). As only some of the targeted compounds were commercially available in certified gas mixtures (isoprene, β-pinene, α-pinene, limonene, p-cymene, myrcene, $\Delta^3$-carene, cis-ocimene, 1.8-cineole, camphor), a gas mixture containing all the compounds reported previously was consequently generated through the vaporization of pure liquid standards inside an electropolished stainless steel canister. Toluene was used as an internal standard to monitor the overall effectiveness of the standard generation.

The BVOC gas mixture was prepared following a three-step procedure:

- A liquid solution of the 20 targeted species was prepared by mixing 100 µL of each individual compound using commercial solutions (Table 2)
- A set-up composed of a heated glass injector connected to an electropolished stainless steel canister (6 L, Silcocan Restek) was used to vaporize and dilute the solution. First, a clean canister was brought under vacuum at $10^{-4}$ bar. A volume of 2 µL of the solution was transferred into the canister through the injector held at atmospheric pressure and ambient temperature. The valve on the canister was opened and the injector was heated at 210°C for 20 minutes. A flow of dry zero air, adjusted at 1 L/min, was then provided to the injector for 18 minutes. Calculated concentrations in the canister ranged from 650 to 750 ppbv for each compound.
- A dilution system composed of three mass flow controllers (MFCs) was used to generate a flow of humid zero air containing 3-4 ppb of each targeted species. Two MFCs were used to generate a flow of zero air (1 L/min) at a relative humidity (RH) of 50% (22°C), the first MFC generating 500 mL/min of dry air and the second MFC connected to a water bubbler generating 500 mL/min of zero air at 100% RH (22°C). The third MFC was used to regulate a flow rate of 5 mL/min for the calibration gas that was mixed with the humid air flow, leading to a dilution factor of 200.

**Table 2 : List of targeted species for ambient measurements and chemical properties for gas standard generation**

| Compounds | Fromula | Molar mass | purity | Supplier |
|---|---|---|---|---|
| Isoprene | $C_5H_8$ | 68.12 | 0.98 | Merck |
| Toluene | $C_7H_8$ | 92.15 | 0.999 | Sigma-Aldrich |
| α-pinene | $C_{10}H_{16}$ | 136.23 | 0.98 | Aldrich |
| Camphene | $C_{10}H_{16}$ | 136.23 | 0.95 | Aldrich |
| Sabinene | $C_{10}H_{16}$ | 136.23 | 0.75 | Sigma-Aldrich |
| β-pinene | $C_{10}H_{16}$ | 136.23 | 0.99 | Aldrich |
| Myrcene | $C_{10}H_{16}$ | 136.23 | 0.7 | TCI |
| 2-Carene | $C_{10}H_{16}$ | 136.23 | 0.97 | Sigma-Aldrich |
| $\Delta^3$-Carene | $C_{10}H_{16}$ | 136.24 | 0.9 | Sigma-Aldrich |
| α-Terpinene | $C_{10}H_{16}$ | 136.23 | 0.87 | ACROS |
| α-Phellandrene | $C_{10}H_{16}$ | 136.23 | 0.65 | TCI |
| p-Cymene | $C_{10}H_{14}$ | 134.22 | 0.95 | TCI |
| Limonene | $C_{10}H_{16}$ | 136.25 | 0.97 | Sigma-Aldrich |
| Ocimene | $C_{10}H_{16}$ | 136.23 | 0.9 | Sigma-Aldrich |
| γ-terpinene | $C_{10}H_{16}$ | 136.23 | 0.95 | ACROS |
| Terpinolene | $C_{10}H_{16}$ | 136.26 | 0.85 | TCI |
| Linalool | $C_{10}H_{18}O$ | 154.25 | 0.97 | Sigma-Aldrich |

| | | | | | | | |
|---|---|---|---|---|---|
| Citral | $C_{10}H_{16}O$ | 152.23 | 0.9 | Sigma-Aldrich |
| Nopinone | $C_9H_{14}O$ | 138.1 | 0.98 | Sigma-Aldrich |
| β-Caryophyllene | $C_{15}H_{24}$ | 204.5 | 0.8 | Sigma-Aldrich |

The stability of the standard mixture in the canister was checked for 2 weeks. The accuracy of the generated concentrations has been evaluated by comparing the response coefficients of three VOCs (toluene, α-pinene and β-pinene) present in the canister to a certified calibration standard of gas mixture (cylinder D09 0523, June 2014, NPL, Table S. 1).

## 2.2 Chromatographic separation and FID detection

Two chromatographic columns recommended by Aerosol, Clouds and Trace Gases Research Infrastructure (ACTRIS), 2014, were tested for separation: BPX-5 (60 m x 0.25 mm i.d., 1 μm thickness; SGE Analytical Science) and DB-624 (60 m x 0.32 mm i.d., 1.80 μm thickness; Agilent J&W). The DB-624 is recommended for terpenes and oxygenated compounds targeted in this study. The BPX-5, which is equivalent to the DB-5 used by Saxton et al. (2007), is recommended for monoterpenes and is more selective than the DB-1 often used for BVOC separation (Table 1) with oxygenated compounds. For both columns, the temperature program, the carrier gas flow rate and the pressure were optimised and final parameters leading to the best separation are shown in Table 3. The detection of each species was made using a flame ionization detector fed by flow rates of 40, 450, and 45 mL/min of $H_2$ (alpha 2, N60), zero air (alpha 1, N50), and $N_2$ (alpha 1, N50), respectively.

**Table 3 : Chromatographic details for BPX-5 and DB-624 columns**

| DB-624 | | | | | | | | BPX-5 | | | | |
|---|---|---|---|---|---|---|---|---|---|---|---|---|
| ΔF (mL/min)/min | Flow (mL/min) | Hold (min) | Run time (min) | ΔT (°C/min) | Temperature (°C) | Hold (min) | Run time (min) | Pressure (PSI) | ΔT (°C/min) | Temperature (°C) | Hold (min) | Run time (min) |
| 0 | 4 | 1 | 1 | 0 | 40 | 8 | 8 | | 0 | 40 | 8 | 8 |
| 0.15 | 2 | 0 | 14.33 | 6 | 135 | 0 | 23.83 | | 6 | 135 | 0 | 23.83 |
| 0.15 | 3 | 10 | 31 | 1.25 | 180 | 0 | 59.83 | 24.3 | 0.6 | 145 | 0 | 40.50 |
| 0.2 | 5 | 1 | 70.67 | 6 | 250 | 3.5 | 75 | | 0 | 250 | 9 | 67 |

## 2.3 Preconcentration and thermodesorption

As mentioned above, the pre-concentration trap was filled with Carbopack B (100mg, Supelco) and was held at 20°C during the sampling step. The sampling flow rate was maintained at 20 mL/min for 60 minutes under field operating conditions, which leads to a sampling volume of 1200 mL. In order to test the volume breakthrough, the sampling duration was varied 6 times from 10 min up to 90 min, leading to sampling volumes ranging from 200 mL up to 1800 mL. This test was carried out by sampling a gas mixture of approximately 12 ppb of each VOC at 80% RH (22°C) using the generation system described in section 2.1. Before desorption, a trap purge was performed with 20 mL of Helium (alpha 1, N50).

The desorption temperature was tested at 275, 300, 325, and 350°C. For each experiment, a mixture of approximately 3-5 ppb of each VOC was generated at 50% RH (22°C) and sampled by the instrument. The desorption was performed twice at the same temperature without additional sampling between the two desorptions, leading to two chromatograms. Two replicates were performed at each temperature. The desorption efficiency was evaluated at each temperature from Eq. (1), using both chromatograms:

$$E_D(\%) = \frac{A_{first_i}}{A_{first_i} + A_{second_i}} \ , \tag{1}$$

Where $E_D$ is the desorption efficiency, $A_{first_i}$ the peak area (a.u.) of the compound $i$ from the first chromatogram, and $A_{second_i}$ the peak area (a.u.) of the same compound $i$ from the second chromatogram.

## 2.4 Ozone Removal

During the pre-concentration step of VOCs on a solid sorbent, unsaturated BVOCs may react with ambient ozone, leading to their loss and the formation of more oxygenated compounds (Lee et al., 2006; McGlenny et al., 1991). These unwanted reactions can be reduced if ozone is selectively removed from the sampled flow before the pre-concentration stage. Three ozone scrubbers were tested. The scrubbers were chosen based on ACTRIS recommendations for BVOC sampling (ACTRIS, 2014): copper tubes coated with potassium iodide (KI) (Helmig, 1997), glass filters impregnated with sodium thiosulfate ($Na_2S_2O_3$) (Plass-Dülmer et al., 2002), and copper screens coated with manganese dioxide ($MnO_2$) (Environnement SA).

Three different tests were performed for each scrubber to quantify (i) the ozone removal efficiency, (ii) losses of BVOCs in the absence of ozone, and (iii) potential ozone-induced losses of BVOCs in the scrubber. The sampling flow rate through the scrubber was adjusted at 1 L/min, leading to residence times of 1, 2 and 0.5 s in the KI, $MnO_2$ and $Na_2S_2O_3$ scrubbers, respectively.

(i)     The scrubbing efficiency was calculated from the ratio of ozone concentrations measured after and before the scrubber. The experimental setup used in the previous experiment was also used here, with the addition of an ozone generator made of a photo-reactor equipped with a mercury lamp in which a flow of 2000 mL/min of zero air (alpha 2, N50) was passed through. An ozone mixing ratio of approximately one ppm was generated in the photo-reactor. Two critical orifices were used to inject a small flow rate of 10-15 mL/min of the ozone mixture into the main flow of zero air (1 L/min) at 50% RH (22°C). The ozone removal efficiency was quantified at 93 ppb for the KI and $MnO_2$ scrubbers and 68 ppb for the thiosulfate one.

(ii)    Losses of BVOCs in the absence of ozone were quantified for each scrubber. A mixture of 2-4 ppb of each VOC (Table 2) was generated using the system described in section 2.1. Three chromatograms were acquired without scrubber. The scrubber was then inserted on the gas generation line (1 L/min) before the sampling point of the GC instrument. First, the generation system was connected and its stability checked by recording 4

chromatograms. Secondly, the scrubber was connected and 4 new chromatograms were monitored. Finally, the scrubber was disconnected to check the stability of the generation system by recording 4 additional chromatograms.

(iii)   Losses of BVOCs in the presence of ozone were also quantified for each scrubber to check for ozone-induced losses inside the scrubbers. Ozone was generated with the setup presented in section 2.2.(i) and mixed within the main flow of 2-4 pbb of BVOCs (1 L/min) at 50% RH (22°C), generated as described in section 2.1. A mixing ratio of 50-100 ppb of $O_3$ was measured in the mixture during these experiments. The chromatograms were acquired using the same sequence as in the previous experiment.

## 2.5 Evaluation of analytical performances

**Concentration determination -** For each compound $i$ the concentration was calculated as follows:

$$C_i = \frac{A_i}{K_i} , \tag{2}$$

Where $A_i$ is the peak area and $K_i$ the response coefficient of the compound $i$ calibrated at a concentration of 3-5 ppb.

**Repeatability -** The measurement repeatability was evaluated from 7 replicates using a mixture of 3-5 ppb of the targeted compounds (Table 2) at 50% RH under laboratory conditions and from 3 replicates using the same mixture under field conditions.

**Memory effect -** The memory effect was also evaluated by recording a chromatogram of zero air right after a chromatogram of the above-mentioned mixture. The memory effect was calculated from Eq. (3):

$$M_i \ (\%) = \frac{A_i}{A_{r_i}} , \tag{3}$$

Here, $A_{r_i}$ is the peak area of a compound $i$ in the VOC mixture and $A_i$ is the peak area of the same compound $i$ present in zero air.

**Linearity -** The linearity was tested from 7.5 to 100 µg· m$^{-3}$ (0.5 to 19.5 ppb for monoterpenes) according to ISO 14662-3 (European Standards, 2015) using a VOC mixture generated at 80% RH. For each compound, the linearity was evaluated using linear regression square coefficient $R^2$ and the maximum relative residuals, as follows:

$$\partial_{max} = max \left( \frac{|C_{regi} - C_{expi}|}{C_{regi}} \right) \times 100 , \tag{4}$$

Here, $C_{regi}$ is the concentration of the compound i calculated from the linear regression at a level i and $C_{expi}$ is the concentration measured at this level i.

***Expanded Uncertainty -*** The measurement uncertainty was calculated for each compound based on a methodology proposed by Hoerger et al. (2015). Some uncertainty terms were added or modified to fit our application.

The combined expanded uncertainty shown in Eq. (5), i.e. $u\chi_{unc}$, includes random errors $u\chi_{prec}$ described by the precision, and systematic errors $u\chi_{ystematic}$.

$$u\chi_{unc}^2 = u\chi_{prec}^2 + u\chi_{systematic}^2 ,$$  (5)

The precision, $u\chi_{prec}$, was calculated as follows:

$$u\chi_{prec}^2 = \left(\frac{1}{\sqrt{3}}DL\right)^2 + \left(\chi * \sigma_{\chi standard}^{rel}\right)^2 ,$$  (6)

Where DL is the detection limit, $\chi$ is the mole fraction of the compound of interest, and $\sigma_{\chi standard}^{rel}$ is the relative standard deviation of replicated measurements of the standard.

Five components were considered as systematic errors: the uncertainty associated to the calibration standard ($u\chi_{cal}$), the systematic error due to the integration (peak overlay or poor baseline separation) ($u\chi_{int}$), the potential artifact due to the

scrubber ($u\chi_{scrub}$), the memory effect ($u\chi_{mem}$) and the linearity ($u\chi_{lin}$) for concentrations above calibration concentration.

$$u\chi_{systematic}^2 = u\chi_{cal}^2 + u\chi_{int}^2 + u\chi_{scrub}^2 + u\chi_{mem}^2 (+u\chi_{lin}^2) ,$$  (7)

The systematic error due to the calibration gas uncertainty was calculated as follows:

$$u\chi_{cal}^2 = \frac{A_{sample}*V_{cal}}{V_{sample}* A_{cal}} * \delta\chi_{cal} ,$$  (8)

where $A_{sample}$ is the peak area of sample measurement, $A_{cal}$ the peak area of the calibration standard measurement, $V_{sample}$ the

sample volume, $V_{cal}$ the sample volume of calibration standard, and $\delta\chi_{cal}$ the uncertainty of the concentration in the standard.

$\delta\chi_{cal}$ depends on the calibration type used for the compound either i) the certified standard NPL or ii) the standard generated as described in section 2.1. In case i), $\delta\chi_{cal}$ is the certified uncertainties given with the certified concentration. In case ii), $\delta\chi_{cal}$ combines an uncertainty fixed by the worst recovery obtained between NPL standard and canister test presented in

section 4.1 and the reproducibility of the generated standard mixture.

The systematic integration error $u\chi_{int}^2$ is defined as :

$$u\chi_{int}^2 = \left(\frac{f_{cal}}{V_{sample}} * \delta A_{sample}\right)^2 + \left(\frac{A_{sample}*V_{cal}*\chi_{cal}}{V_{sample}*A_{cal}^2} * \delta A_{cal}\right)^2 with f_{cal,i} = \frac{V_{cal}*\chi_{cal}}{A_{cal}} ,$$  (9)

Where $\delta A_{cal}$= integration error of the calibration standard measurement, and $\delta A_{sample}$ = integration error of the sample measurement. These integration uncertainties were determined using representative chromatograms.

The systematic error linked to the influence of the scrubber $u\chi_{scrub}^2$ was evaluated as follows:

$$u\chi_{scrub}^2 = \left(\frac{\chi_{free}*e_{scrub}^{rel}}{\sqrt{3}}\right)^2 ,$$  (10)

Where $e_{scrub}^{rel}$ is the relative deviation between measurements of the same mixture without and with scrubber and ozone and $\chi_{free}$ the mole fraction measurement without scrubber and ozone.

The systematic error linked to the memory effect $u\chi_{mem}^2$ is defined as:

$$u\chi^2_{mem} = \left(\frac{M_{i*}\,\chi_{n-1}}{\sqrt{3}}\right)^2 , \tag{11}$$

Where $M_i$ is the memory effect determined following the equation Eq. (3) and $\chi_{n-1}$ is the mole fraction of the previous measurement.

The systematic error $u\chi^2_{lin}$ due to the linearity was calculated as follows:

$$u\chi^2_{lin} = \left(\frac{\chi * \partial_{max}}{\sqrt{3}}\right)^2 , \tag{12}$$

Where $\partial_{max}$ is the maximum relative residuals defined by Eq. (4).

The expanded uncertainty is then estimated as the total uncertainty calculated multiplied by the coverage factor k = 2.

## 2.6 Field measurements: Deployment during the LANDEX field campaign

The online GC-TD-FID instrument was deployed for the first time during the summer 2017 along the LANDEX field campaign. This campaign, whose main objective is to study the formation of secondary organic aerosols (SOAs) in the Landes forest (France), was conducted from 29 June to 19 July 2017. The measurement site was located at Salles-Bilos in the Landes forest (44°29'39.69"N, 0°57'21.75"W, 37 meters above sea level).

A description of the site, which is also part of the European ICOS (Integrated Carbon Observation System) Ecosystem infrastructure, can be found in Moreaux et al. (2011). Briefly, the site consisted of a large area of 30.2 ha (570 × 530 m), mainly composed of maritime pines (*Pinus pinaster*), with a dense understory of gorse (*Ulex europaeus L.*), grass (*Molinia caerulea (L.) Moench*) and heather (*Calluna vulgaris (L.) Hull*). It is part of the Landes of Gascogne forest which has an area of around 1 million ha. The climate is temperate with a maritime influence due to the proximity of the North Atlantic Ocean (25 km). The nearest urban area is the Bordeaux metropole, 50 km northeast from the site.

The mobile laboratory "OMEGA" from IMT Lille Douai was deployed and located between two ranks of trees. Ambient air was sampled through a heated (55°C) 10-m long sampling line (sulfinert, ¼" o.d.) at a flow rate of 1 L/min using an external pump for continuous flushing. The tree height was approximately 10 m and the measurement height was adjusted at 6 m, below the forest canopy. The TD-GC-FID pulled 20 mL/min of air from the sampling line during the pre-concentration steps.

## 3 Method optimization

### 3.1 Chromatographic separation

As mentioned in section 2.2, two chromatographic columns were tested and operating conditions were optimized as shown in Table 3, to obtain the best separation of the 20 targeted BVOCs. Fig.1 illustrates the most difficult compounds to separate. With the DB 624 column, myrcene and β-pinene, two of the main observed monoterpenes, were co-eluted, unlike with the BPX5 column. So, this last column was finally selected for this measurement method.

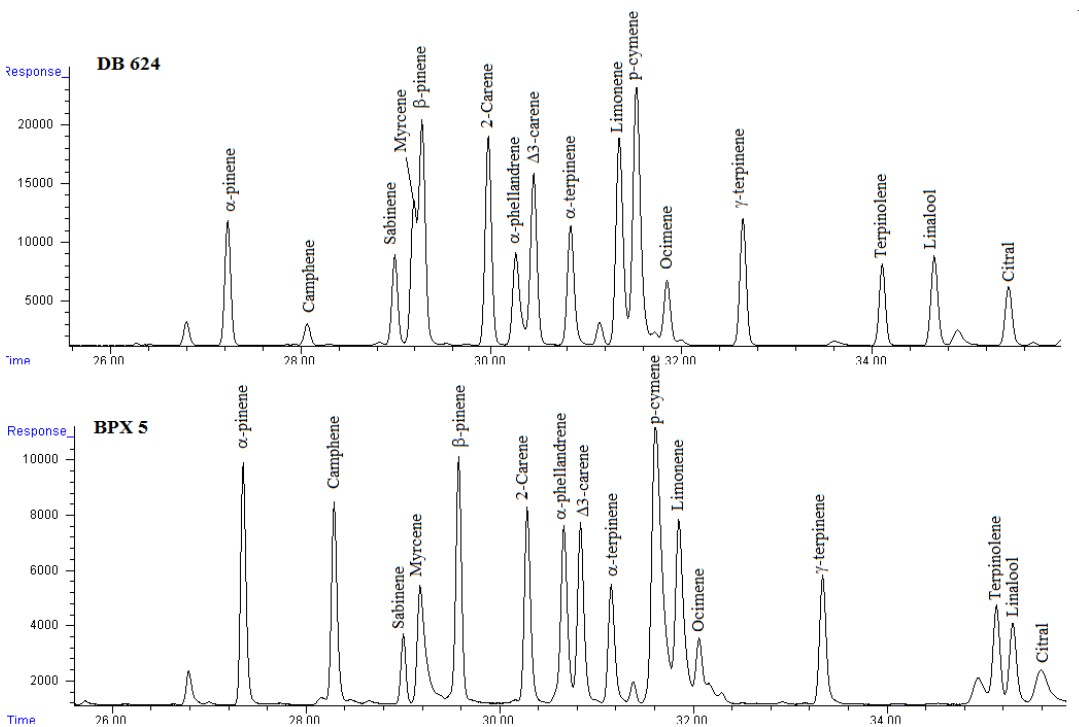

**Figure 1 : Separation of 20 BVOCs from Table 2 using the DB-624 column (top) and BPX5 column (bottom). VOC mixing ratios were approximately 1 ppb.**

The resolution for every compound was at least equal to 1.2, yielding the chromatographic separation to be considered as satisfactory for the 20 targeted BVOCs. Nevertheless, it was also important to consider the potential co-elution with other non-targeted VOCs, such as compounds from potential anthropogenic sources. Twenty selected compounds usually measured in urban areas were added in the test mixture. The separation remained acceptable for most of the compounds except for isobutylbenzene co-eluted with $\Delta^3$-carene, 1,2,3-trimethylbenzene co-eluted with p-cymene and limonene, butylbenzene co-eluted with γ-terpinene, and n-dodecane co-eluted with menthol (Fig. 2). It should be noted that other compounds which have not been targeted here could co-elute with targeted compounds and maybe other monoterpenes.

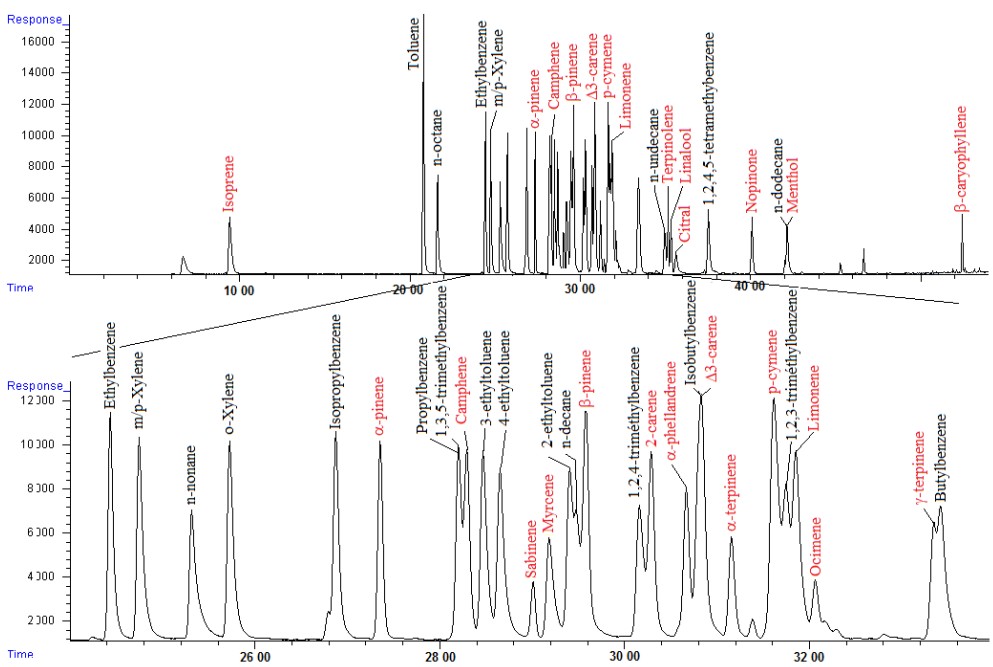

**Figure 2 : Separation of 20 BVOCs together with 20 anthropogenic compounds. VOC mixing ratios were approximately 4 ppb. Compounds written in red character are BVOCs targeted**

### 3.2 Thermodesorption

As mentioned in section 2.3, different desorption temperatures were tested to optimize the quantitative transmission of VOCs inside the GC column. Results reported in Fig. 3-(a) show peak areas observed for each compound after the first thermodesorption, for the 4 different tested temperatures. First, sabine present a lower respond than theoretical expect as monoterpene. The most probable reason is the potential degradation of the sabinene in p-cymene and/or limonene during the thermodesorption, as demonstrated for Tenax and Carboxen by Coeur et al. (1997). For most compounds, the desorption

temperature only has a small influence on the response, with changes lower than 5% between the maximum and minimum peak areas (e.g. toluene, α-pinene, myrcene, camphene, carenes, nopinone). For some compounds, the peak area decreased between 275°C and 350°C, indicating a potential thermodegradation (e.g. -20% for sabinene or -10% for β-pinene and isoprene), consistent with observations made by Hopkins et al. (2011). For other compounds, an increase of the desorption temperature led to an increase of the response, indicating that these compounds may be desorbed more efficiently on

carbopack B (e.g. +20% for β-caryophyllene or α-terpinene).

The desorption efficiency is presented on Fig. 3-(b) for each compound at each temperature. Efficiency was higher than 95% for all compounds except for citral and β-caryophyllene, with an efficiency increase of 22% for both compounds when temperature raised from 275°C to 350°C. The gain between 325°C and 350°C was of 1-2%. From these results, a temperature of 325°C was considered as a good compromise between losses of thermosensitive compounds and desorption

efficiency for less volatile compounds.

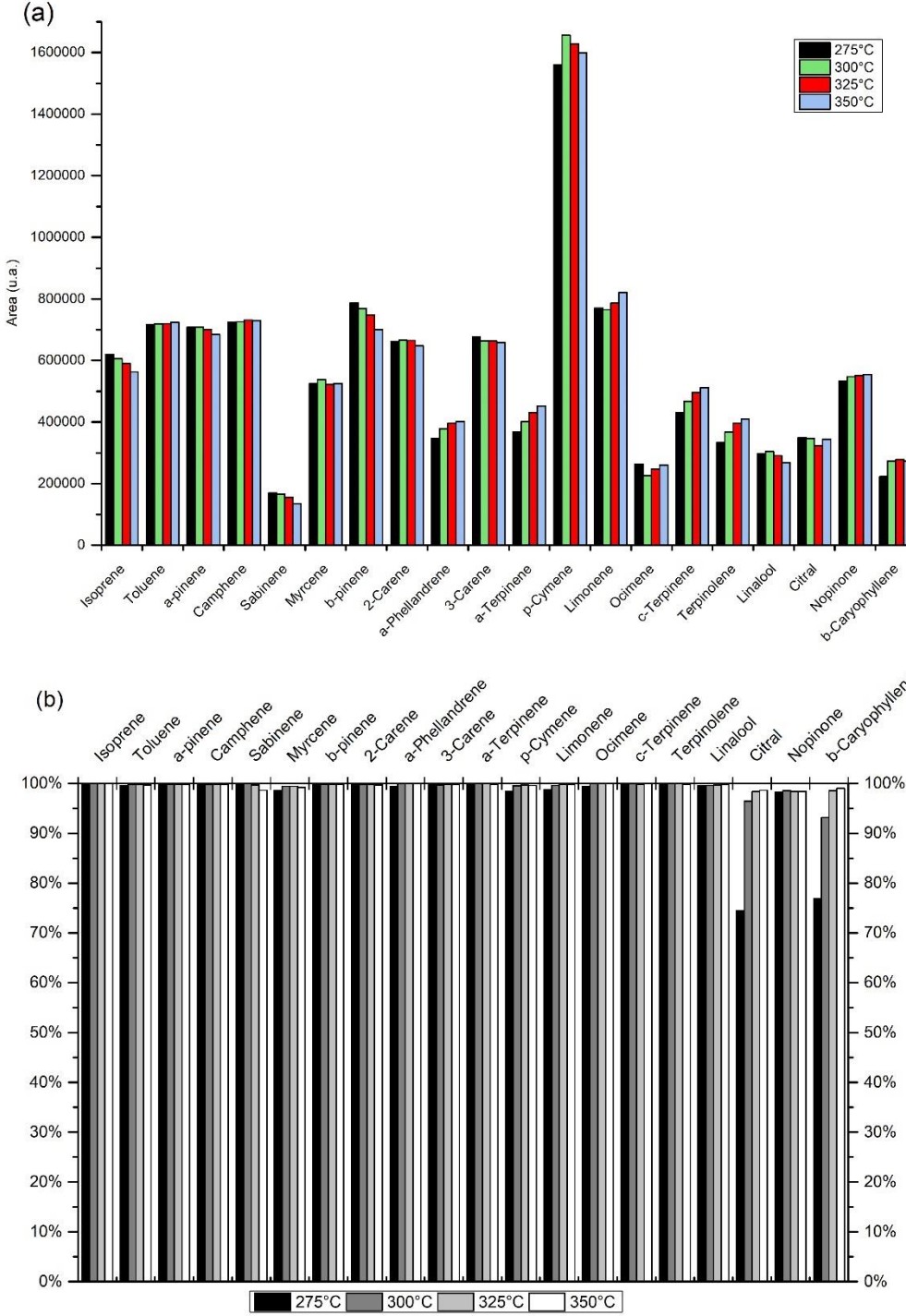

**Figure 3 : Investigation of the optimal desorption temperature for BVOC measurements. 2 replicates performed at each temperature (a) peaks area for the first desorption analysis and (b) desorption efficiency**

**3.3 Safe sampling volume**

The term breakthrough is defined as the volume of gas that causes a compound migration through 1 g of the adsorbent bed at a specific temperature. The breakthrough volume enables the estimation of the maximum sampling volume that ensures a quantitative sampling of a compound using a certain adsorbent mass at a specific temperature (Dettmer and Engewald, 2002). The most volatile compound will be the first to breakthrough from the trap. Several experiments were performed to evaluate the safe sampling volume of the trap. The VOC mixture was generated as presented in section 2.1 at a concentration of 8-21.7 ppb at 80%RH (22°C). Six volumes have been tested: 200, 600, 1000, 1200, 1440 and 1800 mL. The peak area observed on the chromatograms for each compound was plotted versus the sampled volume as shown in Fig. 4 for the four most volatile species (isoprene, toluene, α-pinene and camphene). A linear increase of the peak area with the sampling volume is observed on Fig. 4, indicating that the compounds were quantitatively retained on the sorbent and the breakthrough was not reached up to a sampling volume of 1800 mL. To be conservative, the sampling volume was set at 1200 mL for this GC instrument.

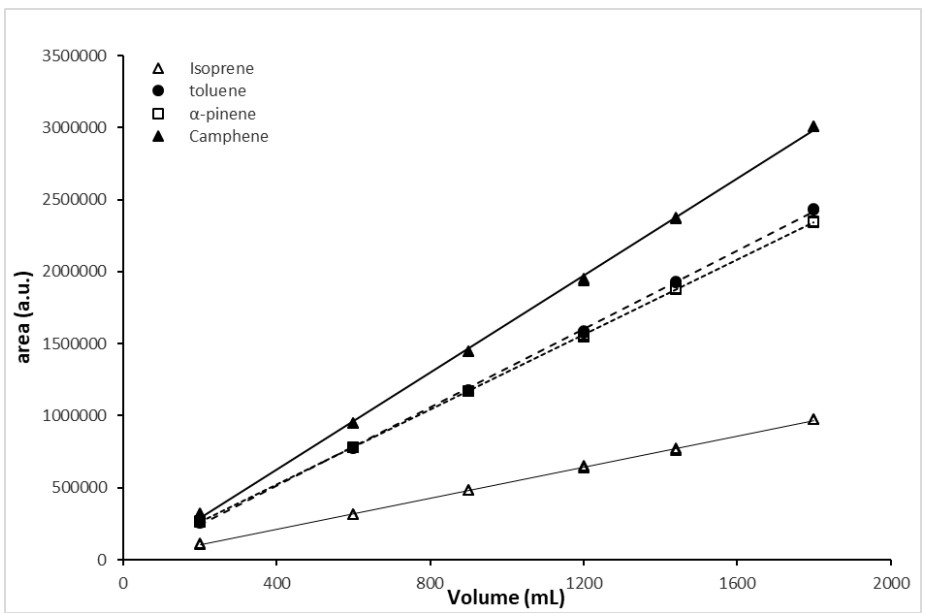

**Figure 4 : Investigation of the safe sampling volume at 80% RH (22°C) for the four most volatile compounds in the mixture**

**3.4 Comparison of ozone scrubbers**

Several experiments were performed to test the impact of ozone on the measurements. The results reported in Fig. 5 indicate a significant loss for a large number of reactive compounds in the absence of an ozone scrubber, including myrcene (-15%), sabinene (-27%), γ-terpinene (-38%), ocimene (-32%), linalool (-46%), α-phellandrene (-48%), terpinolene (-55%), α-terpinene (-82%) and β-caryophyllene (-99.5%). In addition, it is clear that oxidation products generated inside the trap also impacted the measurements of citral, p-cymene and α-pinene, likely due to coelutions. These results stress the need of using

an ozone scrubber to reduce any measurement bias involving the reaction of $O_3$ with adsorbed BVOCs in the trap when a pre-concentration technique is used.

The measured output-to-input ozone ratio indicated a removal efficiency better than 99.4% for each scrubber. The $MnO_2$ scrubber is a commercialized version used in the ozone monitor, with a recommended lifetime of one year. The KI scrubber is efficient during at least one month. However, it has to be replaced more frequently in case of sampling under high humidity conditions (EPA, 1999). The thiosulfate scrubber is efficient for approximately 16 hours (Plass-Dülmer et al., 2002).

Losses of BVOCs in each scrubber were evaluated using the procedure described in section 2.4 and are reported in Fig. 5. During these tests, the $MnO_2$ scrubber exhibited losses of oxygenated compounds ranging from 40% to 80% (see supplementary Fig. S1) while the two other scrubbers did not exhibit losses larger than 5-10%. This observation is consistent with cautionary remarks reported for $MnO_2$ scrubbers by ACTRIS (2014). This scrubber was therefore rejected for our application. The thiosulfate and KI scrubbers exhibited losses lower than 5% for most of non-oxygenated BVOCs.

The thiosulfate and KI scrubbers were tested in the presence of ozone to check for ozone-induced losses inside the scrubber. Results presented in Fig. 5 indicate that significant losses were observed only for the two most reactive compounds, i.e. α-terpinene and β-caryophyllene, with losses of approximately 20% and 40% respectively. For other compounds, this loss was always lower than 15%, which is reasonable for ambient measurements. Since both scrubbers exhibited similar results, the KI scrubber was selected based on its longer lifetime, limiting the number of measurement interruptions along the field campaign.

In order to propose an exhaustive overview of ozone scrubber choice for BVOC measurements, a comparison of our results with a KI scrubber compared to those obtained by Hellen et al. (2012) with a heated stainless steel tube of 3 m length (SS 3 m), at a flow rate of 1 L/min and with or without 50 ppb of ozone, have been realised. Without ozone, the recovery results with both types of scrubbers are comparable for toluene, nopinone, and monoterpenes (94-97%), except for terpinolene and camphene. β-caryophyllene and terpinolene recoveries are slightly better with the SS 3 m (103% and 104% respectively) than with the KI scrubber (98% and 95% respectively). Linalool and camphene recoveries are slightly better using the KI scrubber (93% and 96% respectively) than with the SS 3 m (87% and 91% respectively). With ozone, the monoterpenes, β-caryophyllene and nopinone recoveries are comparable or slightly better with SS 3 m than with the KI scrubber (97-110%). Linalool presents a bad recovery with the SS 3 m of 54% compared to 89% with the KI scrubber. Here, we compared our results to the results of a SS 3 m but we used a longest tube during the campaign, more like the SS 5 m length presented by Hellen et al. (2012). The recoveries of β-pinene, linalool and β-caryophyllene with a SS 5 m and no ozone are smaller than with a SS 3 m with no ozone. As stated by Hellen et al. (2012), the compound isomerization might be the reason for this. β-pinene is known to isomerize easily in myrcene and limonene during the heating step. For all those reasons, we finally preferred the use of a KI scrubber in our study.

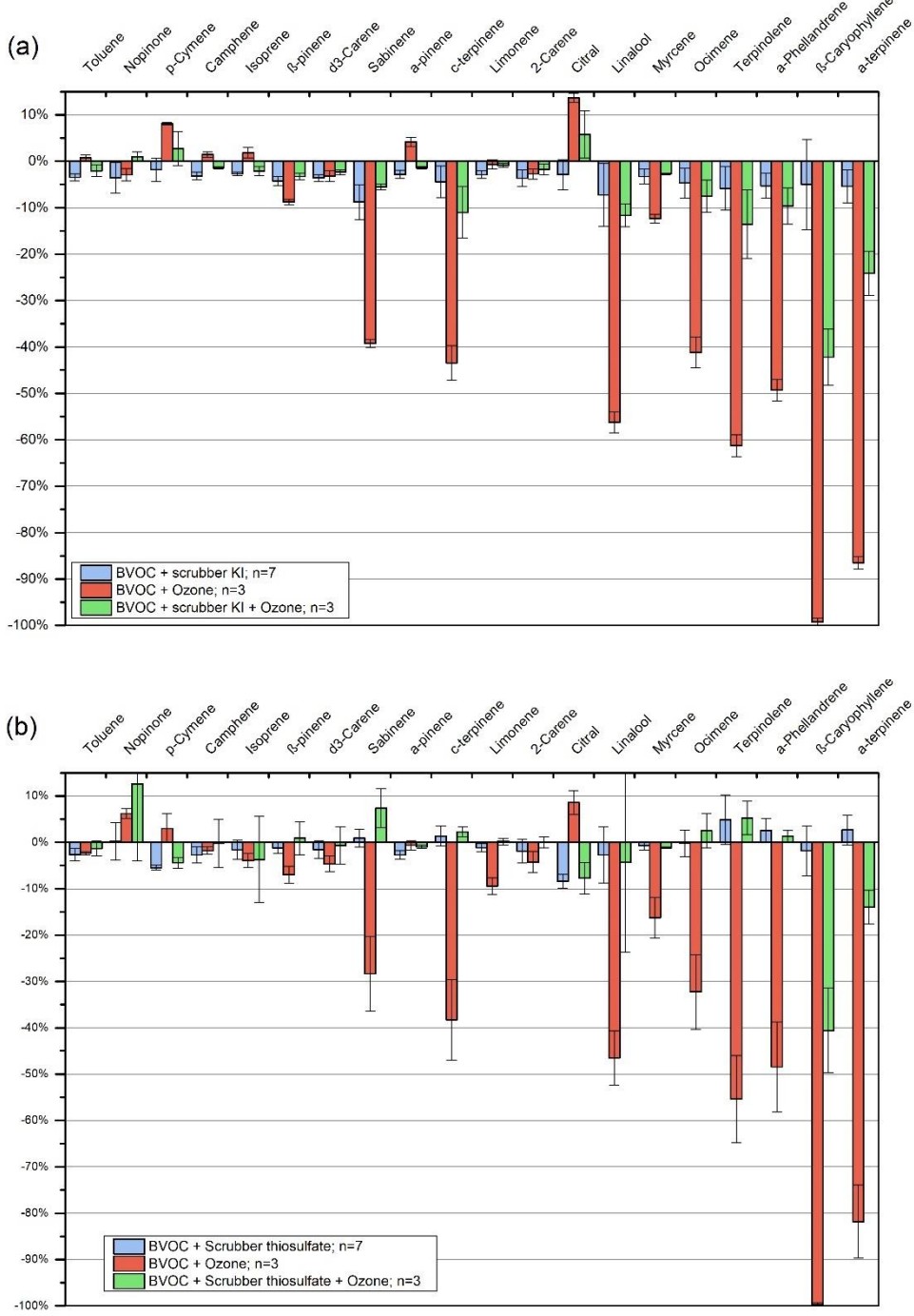

**Figure 5 : Relative deviation to BVOC measurements performed (i) with scrubber and without O$_3$, (ii) when O$_3$ (50 ppb) is added in the standard mixture without scrubber and (iii) when a scrubber is used with O$_3$. (a): KI scrubber and (b): thiosulfate scrubber. VOCs are classified from the less reactive (left) to the most reactive (right) with ozone. Errors bars correspond to 3 standard deviation values.**

**3.5 Optimized method**

Resulting from the tests performed, the optimized method for BVOCs measurement is:

*Sampling:* the online TD system sampled on a trap made of Carbopack B at a temperature of 20°C and a flow rate of 20 mL/min during 60 min. For the desorption, the trap was quickly heated from 20°C to 325°C and maintained at 325°C during 15 minutes with a helium flow rate of 20 mL/min. The transfer line between the TD and GC was maintained at 140°C.

*Analysis:* the GC system was equipped with a chromatographic BPX5 column (60 m × 0.25 mm i.d. and 1 μm film thickness; SGE Analytical Science). Pressure was maintained at 24 psi. The temperature followed these settings: $T_{oven}$ (initial): 40 °C for 8 min, $T_{oven}$ (first ramping): 6°C/min rate until 135°C, $T_{oven}$ (second ramping): 0.6 °C/min rate until 145°C, $T_{oven}$ (third ramping): 6°C/min rate and $T_{oven}$ (final): 250°C for 5 min. The FID detector was fed by pure $H_2$ = 40 mL/min, pure air = 450 mL/min and pure $N_2$ = 45 mL/min ($T_{FID}$ = 250°C).

**4 Analytical performances results**

**4.1 BVOC calibration**

Fig. 6 presents the Relative Deviation (RD) between response coefficients determined from the mixture generated from a canister, as described in section 2.1, and the response coefficients determined from the NPL standard for toluene, α-pinene, and β-pinene. The RD was lower than 5% for toluene while a systematic value of -10% was obtained for α-pinene, indicating higher values measured with the NPL standard than with the generated mixture. RD reached +32% for β-pinene, this factor being further used for the calculation of $\delta\chi_{cal}$ for uncertainty evaluation. Note that some instability of conservation was previously demonstrated for β-pinene in a high pressure cylinder by Rhoderick and Lin (2013). Therefore, a doubt was emitted on the NPL certified standard value for β-pinene. Concentration of all compounds was stable during 2 weeks. Relative standard deviations (RSD) are reported in Table 4 for each compound and for 7 canisters filled with the generated standard mixture. For monoterpenes, reproducibility ranged from 3.5% to 9.8%, most of them being below 5%. For oxygenated monoterpenes and nopinone, reproducibility was between 9% and 11% and raised up to 22% for β-caryophyllene. Although significant bias was observed with the NPL standard for β-pinene, the reproducibility of the mixture generation was considered to be satisfactory.

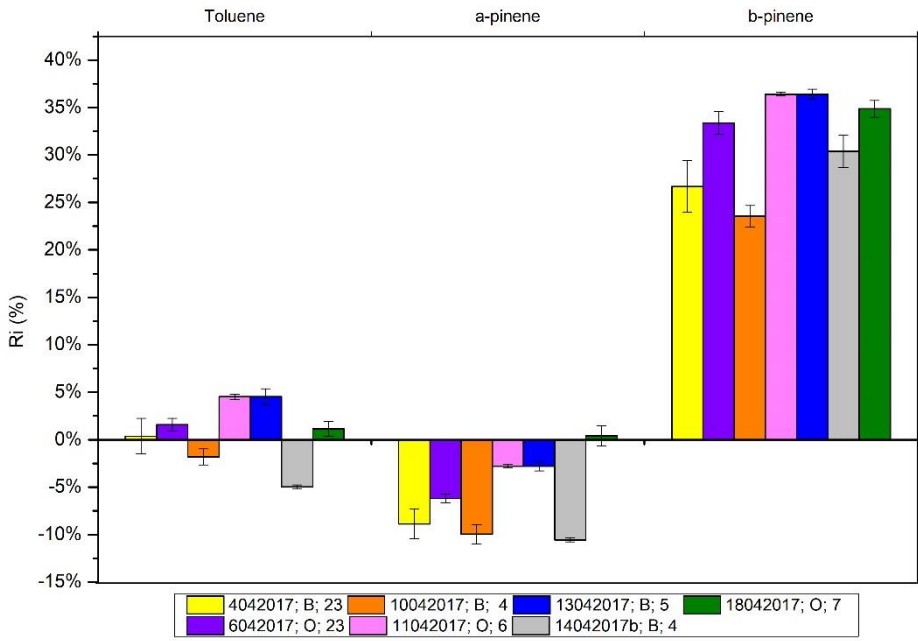

**Figure 6 : Canisters respond coefficient relative deviation to the NPL respond coefficient. Legend: canister creation date ddmmyyyy; canister used B: S053 and O: S052; n= number of replicates. Error bars = 1σ.**

**Table 4 : Canister reproducibility (n=7)**

| Compounds | Concentration (ppb) | RSD (%) |
|---|---|---|
| Isoprene | 1880.4 | 5.1% |
| Toluene | 1244.9 | 3.1% |
| α-pinene | 815.3 | 4.1% |
| Camphene | 842.4 | 4.8% |
| Sabinene | 626.2 | 4.5% |
| Myrcene | 567.7 | 5.2% |
| β-pinene | 828.6 | 3.5% |
| 2-Carene | 812.8 | 4.9% |
| α-Phellandrene | 533.0 | 5.8% |
| 3-Carene | 745.1 | 4.2% |
| α-Terpinene | 713.4 | 3.5% |
| p-Cymene | 830.0 | 6.6% |
| Limonene | 820.4 | 4.9% |
| Ocimene | 717.4 | 9.8% |
| γ-Terpinene | 784.7 | 4.1% |

| | | | | |
|---|---|---|---|---|
| Terpinolene | 729.9 | 5.2% | | |
| Linalool | 642.3 | 11.0% | | |
| Citral | 689.3 | 9.3% | | |
| Nopinone | 874.6 | 9.3% | | |
| β-Caryophyllene | 452.7 | 21.6% | | |

## 4.2 Linearity, repeatability, limit of detection and memory effect

The $R^2$ values from the scatter plot of the instrument response versus compound concentration were higher than 0.99 for all the compounds, except for menthol (0.954) (Table 5). The maximum relative residuals was less than 30% for all compound except for isoprene (44.2%) and menthol (95.6%). For all compounds, measurements were linear between the detection limit

(DL) and 100 µg. m$^{-3}$ (19.5 pbb for monoterpenes). For menthol, measurement was linear between DL and 73.8 µg. m$^{-3}$ (11.5 ppb) ($\partial_{max}$ = 13.8%; slope = 109 203 and $R^2$ = 0.9903).

The relative standard deviation (RSD) has been evaluated for each compound's peak. RSD results are reported in Table 5. RSD were lower than 3% for all compounds, under the laboratory conditions. Repeatability was slightly degraded for some compounds under field conditions. This was logically expected due to environmental change impacting working conditions.

Nevertheless, RSD was lower than 4% except for β-caryophyllene (6.9%) and β-pinene (5.8%). Detections limits were determined for each compound as 3 times the signal to noise ratio value. As presented in Table 5, DL ranged from 5 to 19 ppt and are comparable to those reported in previous studies (Hopkins et al., 2011; Jones et al., 2014; Pankow et al., 2012). The memory effect was reported in Table 5 and always lower than 5% for all compounds which corresponds to the criteria given in the ISO 14662-3 European Standards (2015).

**Table 5 : Concentrations, relative standard deviations (RSD), memory effect, detection limits (DL) and concentrations of maximal relative residual, $\partial_{max}$ (in %), measurement at 80%RH (22°C) for the targeted compounds**

| Compounds | Concentration (ppb) | RSD (%) | | Memory effect (%) | DL (ppt) | conc. $\partial_{max}$ (ppb) | $\partial_{max}$ |
|---|---|---|---|---|---|---|---|
| | | Laboratory | Field | | | | |
| Isoprene | 4.5 | 2.2% | 2.2% | 1.5% | 12 | 1.3 | 44.2 |
| Toluene | 5.9 | 2.1% | | 0.5% | 10 | 3.5 | 5.7 |
| α-pinene | 4.0 | 2.0% | 1.4% | 4.3% | 4 | 1.2 | 12.0 |
| Camphene | 5.0 | 2.2% | 0.8% | 0.3% | 5 | 1.5 | 11.1 |
| Sabinene | 3.2 | 1.7% | 3.0% | 2.9% | 19 | 1.0 | 15.0 |
| Myrcene | 2.8 | 2.1% | 0.5% | 0.7% | 6 | 0.8 | 13.0 |
| β-pinene | 3.8 | 1.9% | 5.8% | 0.5% | 6 | 1.2 | 18.5 |
| 2-Carene | 3.8 | 2.3% | 1.1% | 0.6% | 5 | 1.1 | 9.7 |
| α-Phellandrene | 2.5 | 2.0% | 3.1% | 0.4% | 6 | 0.8 | 3.4 |
| $\Delta^3$-Carene | 3.5 | 2.0% | 3.5% | 0.2% | 5 | 1.0 | 11.3 |
| α-Terpinene | 3.5 | 2.4% | 2.2% | 0.2% | 8 | 1.0 | 30.0 |

| | | | | | | | |
|---|---|---|---|---|---|---|---|
| p-Cymene | 3.8 | 2.0% | 3.7% | 0.8% | 14 | 1.1 | 29.3 |
| Limonene | 3.7 | 2.2% | 1.3% | 0.3% | 4 | 1.1 | 14.5 |
| Ocimene | 3.3 | 2.8% | 2.7% | 0.2% | 4 | 1.0 | 26.9 |
| γ-Terpinene | 3.6 | 2.1% | 2.7% | 0.3% | 8 | 1.1 | 15.9 |
| Terpinolene | 3.5 | 2.1% | 2.9% | 1.5% | 9 | 1.0 | 16.8 |
| Linalool | 3.6 | 1.2% | 3.7% | 2.1% | 11 | 1.0 | 23.9 |
| Citral | 3.0 | 1.1% | 4.0% | 0.4% | 8 | 0.9 | 26.3 |
| Eucalyptol | 2.0 | - | 0.5% | - | 10 | - | - |
| Menthol | 4.1 | 1.5% | - | 2.4% | 9 | 1.0 | 95.6 |
| Nopinone | 4.7 | 2.3% | 3.9% | 2.9% | 7 | 1.5 | 27.3 |
| β-Caryophyllene | 2.1 | 1.1% | 6.9% | 0.6% | 9 | 3.2 | 14.6 |

## 4.3 Measurement uncertainties

The uncertainties compile the analytical performances presented above. Results presented in Fig. 7 have been determined at two different mixing ratios: 2 ppb (Fig. 7-(a)) and 100 ppt (Fig. 7-(b)). The relative part of each uncertainty component was reported as a percentage of the expanded uncertainty at mean value for all compounds. For α-pinene and isoprene, the major component of uncertainty was the precision (e.g. repeatability and DL) representing more than 66% of the variance. For β-pinene, the uncertainties due to the calibration and the scrubber were the most significant. For limonene, the integration was the major one. For the 4 compounds presented in the NPL standard (i.e. isoprene, α-pinene, β-pinene and limonene) the uncertainties ranged from 5% to 10% at 2 ppb and from 7 ppt to 15 ppt at 100 ppt. These results indicated that for these compounds, the method almost complied with the very strict ACTRIS Data Quality Objectives (DQOs) which are 10% above 100 ppt and less than 10 ppt below, and meet the GAW criteria which are 20% above 100 ppt and less than 20 ppt. For the 16 other compounds calibrated with the generated mixture, the major uncertainty component was the calibration factor and their uncertainties ranged from 47% to 99%. In accordance with results presented in the paragraph 3.1, the uncertainties related to α-terpinene, β-caryophyllene, α-phellandrene, γ-terpinene and terpinolene were due to the scrubber. The integration component was significant for p-cymene, limonene, citral and linalool, as expected considering the separation presented on Fig. 2. For sabinene, p-cymene and isoprene, the weight of precision factor was important due to their elevated DL.

During the campaign, another NPL certified calibration standard containing additional monoterpenes such as $\Delta^3$-carene, myrcene with certified uncertainties less than ±5% and p-cymene and ocimene at ±20% was used. Consequently, uncertainties of $\Delta^3$-carene and myrcene decreased to comparable level (±10%) to limonene (Fig. S2). For p-cymene and ocimene at 2 ppb for which the calibration component is major, their uncertainties decreased drastically from 68% and 96% to 25% and 23%, respectively. Integration became the most important source of uncertainty for p-cymene and ocimene. These results point out the need of certified gaseous standard for the measured monoterpenes to reach the most demanding data quality objectives.

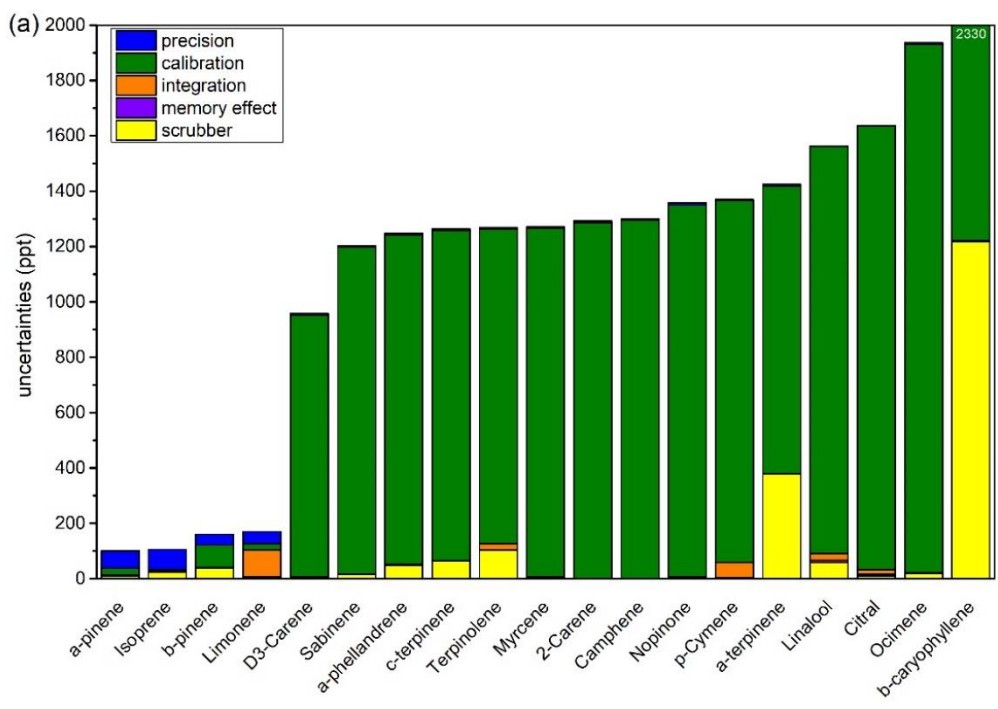

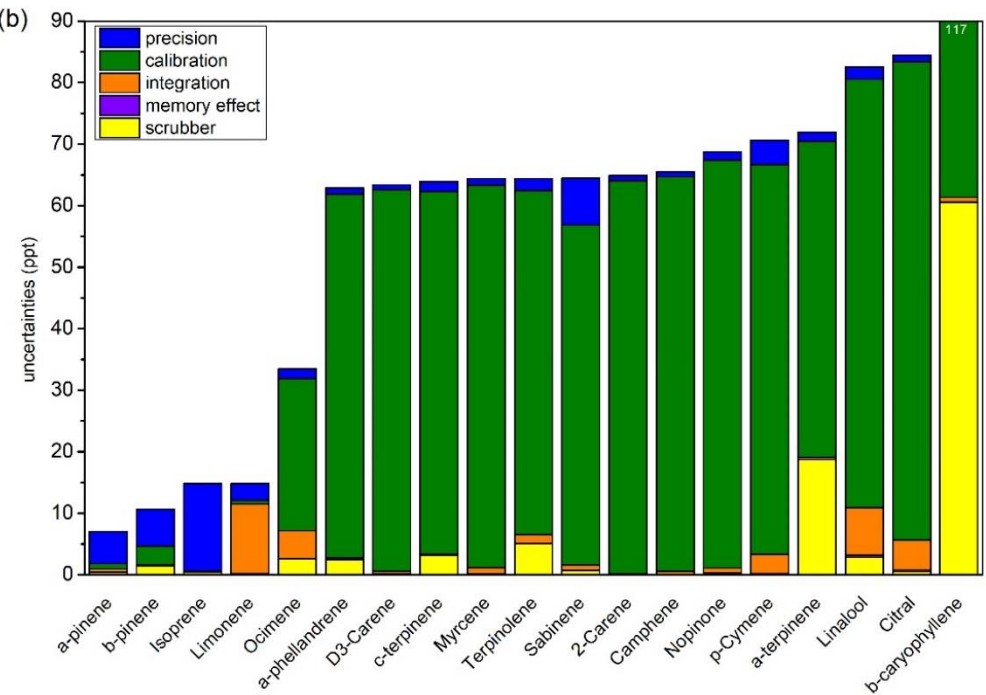

**Figure 7 : Uncertainties repartition for the five terms considered (precision, calibration, integration, memory effect and scrubber) at (a) 2 ppb and (b) 100 ppt with laboratory analytical parameters.**

## 5 Field measurements

The Table 6 presents an overview of the whole dataset acquired specifically during the LANDEX field campaign. The data validation rate was greater than 72% over the 27 days of the campaign. The two major monoterpenes observed were β-pinene and α-pinene, representing on average 60% of the terpenoids measured while isoprene represented about 17%.

**Table 6 : Statistics and uncertainties calculated at the mean value for BVOCs measured during the LANDEX field campaign in July 2017. The calibration technique used depending on the compounds is defined as (A) for the certified NPL gas standard and (B) for the generated mixture**

| Compounds | Min (ppt) | Max (ppt) | Moyen (ppt) | Incertitudes (ppt) | Calibration system used |
|---|---|---|---|---|---|
| β-pinene | 89 | 9902 | 1153 | 153 | A |
| α-pinene | 157 | 8928 | 1138 | 52 | A |
| Isoprene | <DL | 3616 | 408 | 26 | A |
| Myrcene | 20 | 1006 | 147 | 13 | A |
| 3-Carene | 16 | 1191 | 141 | 14 | A |
| Limonene | 9 | 1107 | 138 | 16 | A |
| p-Cymene | <DL | 842 | 124 | 35 | A (±20%) |
| Camphene | 17 | 924 | 120 | 79 | B |
| Linalool | <DL | 554 | 93 | 74 | B |
| Citral | 9 | 660 | 82 | 70 | B |
| Nopinone | <DL | 750 | 62 | 43 | B |
| Eucalyptol | <DL | 255 | 51 | 12 | A |
| Sabinene | <DL | 183 | 42 | 34 | B |
| β-Caryophyllene | <DL | 294 | 36 | 43 | B |
| Ocimene | <DL | 225 | 26 | 8 | A (±20%) |
| Terpinolene | <DL | 126 | 22 | 18 | B |
| α-Terpinene | <DL | 26 | 12 | 14 | B |
| γ-Terpinene | <DL | 41 | 10 | 12 | B |
| α-Phellandrene | <DL | 42 | 9 | 10 | B |
| 2-Carene | <DL | 8 | <DL | 8 | B |

In order to give an insight of the performance of the method, the time series of the uncertainty apportionment is plotted for β-pinene on Fig. 8 for the whole campaign. As reported on section 3.3, β-pinene uncertainty was driven by the precision term, but depending on the period, the integration term became significant due to asymmetric peaks. The linearity term had also an important influence for mixing ratio higher than the calibration values.

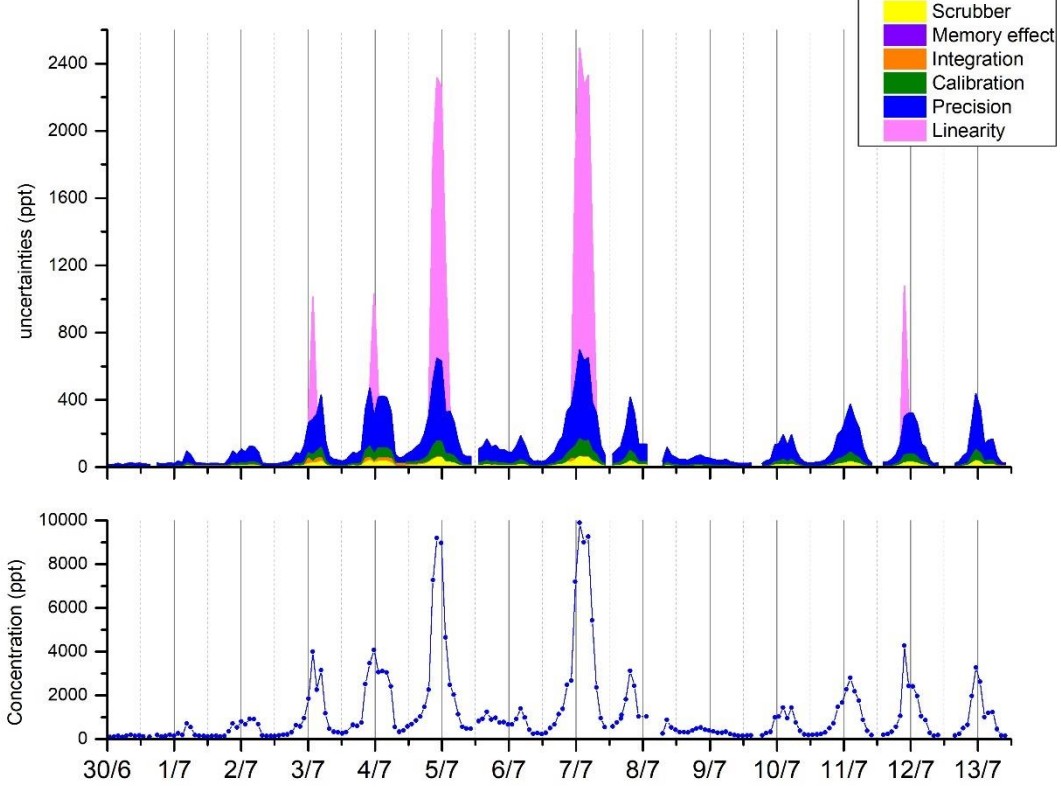

**Figure 8 : Uncertainty (k=2) repartition for the 6 terms considered (precision, calibration, integration, memory effect, scrubber and linearity) (upper panel) and concentration of β-pinene during the Landex-episode 1 field campaign (lower panel).**

To further evidence the high quality of the measurements performed during the campaign, expanded uncertainties were determined at mean value for each BVOC. They were below 13% for the 6 most abundant terpenes, with an excellent value of 4.6% for α-pinene. For the 6 less abundant BVOCs, uncertainties ranged between 31 to 160%, due to concentration levels near the DL. The 7 others measured BVOCs presented uncertainties between 23.5 and 60%, which still allowed to observe significant concentration variations during the campaign (Fig 9 and Fig. S3).

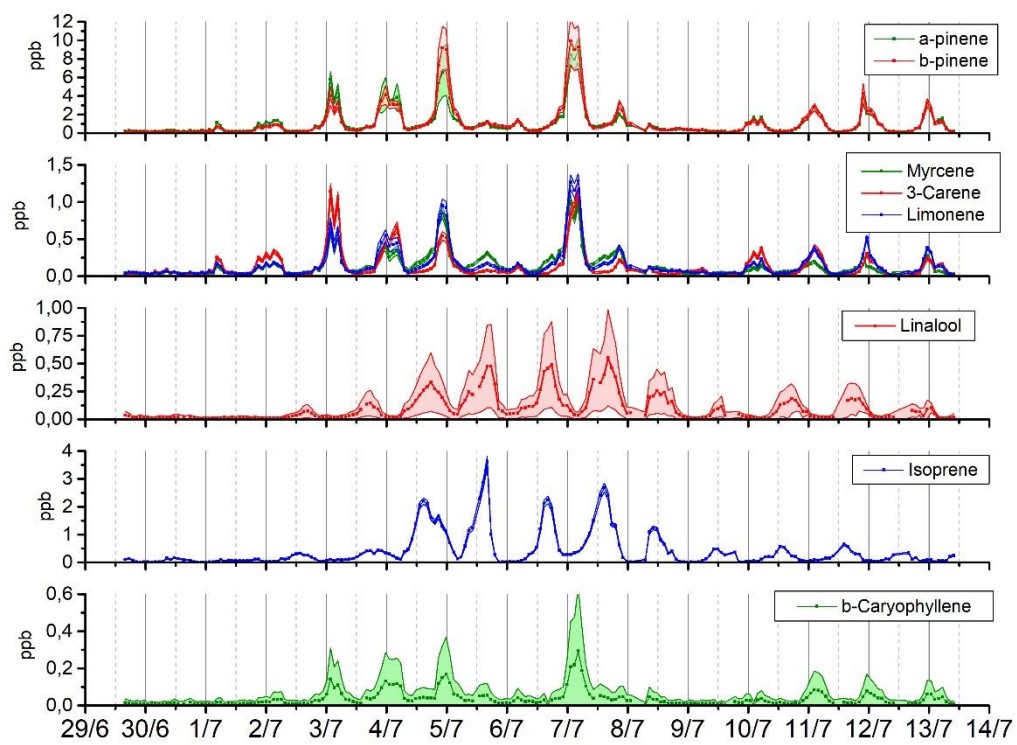

**Figure 9 : Times series of concentrations with their associated uncertainties (k=2) for a selection of BVOCs observed during the LANDEX campaign (β-pinene, α-pinene, limonene, myrcene, Δ³-carene, linalool, isoprene and β-caryophyllene).**

Strong variations in the concentrations were observed for most compounds. α-pinene, β-pinene, myrcene, Δ³-carene,
limonene, p-cymene, camphene, citral, eucalytpol, ocimene, terpinolene, γ-terpinene, α-phellandrene and β-caryophyllene
showed the same pattern with a nocturnal maximum between 22:00 TU and 6:00 TU. For linalool and isoprene, the daily
profile presents a maximum between 10:00 TU and 20:00 TU. These results are consistent with previous observations of the
most abundant monoterpenes at this site (Kammer, 2016; Kammer et al., 2018) in terms of concentrations and daily
variations. Although this work allowed us to monitor for the first time at this site 20 selected BVOCs with a time resolution
of 90 min and hence to provide a highly speciated BVOCs, their composition and time variations will be investigated in
details in a next work in order to assess their contribution to SOA formation in the Landes forest.

**6 Conclusion**

An automated method based on thermal desorption coupled to GC-FID for the online ambient measurement of 20 BVOCs
with a 90 min time resolution was successfully developed and optimized. The analytical performances were satisfying for
ambient measurements. Detection limit ranging from 4 ppt for α-pinene to 19 ppt for sabinene were obtained for a sampling

volume of 1200 mL. Good repeatability was obtained with a relative standard deviation below 5% and a memory effect of less than 2% for all compounds. Uncertainties have been calculated and were below 15% for the six major terpenes. The other compounds presented relative uncertainties between 23.5% and 110% except for 2-carene (> 160%). The major source of uncertainty was the calibration, stressing the need of certified gaseous standards for a wider panel of BVOCs.

The first measurements with the developed method were carried out during the LANDEX-Episode 1 field campaign in summer 2017 at the site of Salles-Bilos. The 3-weeks field measurements demonstrated the excellent performance of the methodology to provide speciated BVOC concentration measurements to further investigate atmospheric BVOC reactivity. β-pinene and α-pinene are the most abundant monoterpenes with concentrations ranged between 89 ppt and 9.9 ppb and between 157 ppt and 8.9 ppb, respectively.

**Data available**

Data will be available on AERIS website on atmospheric section: www.aeris.fr

**Supplement link**

**Author contribution**

KM with the support of TL carried all the experiment out. SS and TS validated uncertainties calculations and supported KM to carry them out. PMF was the logistical support during Landex campaign. SD, NL, EP and EV were supervising the study. KM prepared the manuscript with contributions from all co-authors.

**Competing interests**

The authors declare that they have no conflict of interest.

**Acknowledgements**

This study was supported by the French Environment and Energy Management Agency (ADEME), the CNRS-INSU LEFE-CHAT and the LabEx COTE.

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

**Table 7 : Comparison of operated conditions (sampling method, detector and column used) and limit of detection (DL) (number of species follow)**

| | This study | Hakola et al. (2017) | Jones et al. (2014) | Pankow et al. (2012) | Hopkins et al. (2011) | Hakola et al. (2006) | Greenberg et al. (2004) |
|---|---|---|---|---|---|---|---|
| On-line/not | on-line | on-line | on-line | not on-line | on-line | not on-line | not online |
| Collection | ATD, 1.2 liter | ATD, 1 liter | ATD, 0.75 liter | ATD, 5 liter | ATD, 1 liter | ATD, 3 liter | ATD, 6 liter |
| composition trap | Carbopack B | Tenax TA Carbopack B | Tenax | Tenax-TA and Carbotrap B or Tenax GR and Carbograph | Carboxen 1000 and Carbotrap B (90 mg) | Tenax-TA carbopack-B | glass beads (80 mg), Carbotrap B (170 mg), Carbosieve III (350 mg) |
| Detection | GC/FID | TDGC-MS | GC/FID | GCxGC ToFMS | GCxGC FID | GC/MS | GC/ MS |
| Column dimension | BPX-5 60 m, 0.25 mm i.d.,1 µm | DB-1 60 m, 0.25 mm i.d., 0.25 µm | MXT-5 15m, 0.,25mm i.d., 0.25µm | DB-VRX, Stabilwax 45m,0.25mm id, .,4µm; 1.5m, 0.25mm id, 0.25µm | PLOT, LOWOX 50m,0.53mm id; 2x10m, 0.3mm id | HP-1 60 m, 0.25 mm i.d. | DB-1 30 m, 0.32 mm i.d., 1 µm |
| LoD definition | Signal/noise (S/N) = 3 | not stated | not stated | S/N = 10 | not stated | not stated | not stated |
| compound | pptv | pptv | pptv | pptv | pptv | pptv | pptv |
| Isoprene | 12 | | | 4 | 1 | 11.4 | 1 |
| monoterpenes | 4-19 (14) | <1 (8) | 4-5 (12) | 0.7-2.1 (9) | 3 (5) | 5.2-10.7 (7) | 1 (4) |
| Oxygenated monterpenes | 4-11 (4) | | 4 (4) | 1.3-1.9 (6) | | 13.2 (1) | |
| Oxidation product | 7 (1) | <1 (1) | 5 (2) | 2.1 (1) | | | |
| sesquiterpenes | 9 (1) | <1 (6) | | 0.9-1.4 (4) | | 9.4 (1) | |

**Table 8 : List of targeted species for ambient measurements and chemical properties for gas standard generation**

| Compounds | Fromula | Molar mass | purity | Supplier |
|---|---|---|---|---|
| Isoprene | $C_5H_8$ | 68.12 | 0.98 | Merck |
| Toluene | $C_7H_8$ | 92.15 | 0.999 | Sigma-Aldrich |
| α-pinene | $C_{10}H_{16}$ | 136.23 | 0.98 | Aldrich |
| Camphene | $C_{10}H_{16}$ | 136.23 | 0.95 | Aldrich |
| Sabinene | $C_{10}H_{16}$ | 136.23 | 0.75 | Sigma-Aldrich |
| β-pinene | $C_{10}H_{16}$ | 136.23 | 0.99 | Aldrich |
| Myrcene | $C_{10}H_{16}$ | 136.23 | 0.7 | TCI |
| 2-Carene | $C_{10}H_{16}$ | 136.23 | 0.97 | Sigma-Aldrich |
| $\Delta^3$-Carene | $C_{10}H_{16}$ | 136.24 | 0.9 | Sigma-Aldrich |
| α-Terpinene | $C_{10}H_{16}$ | 136.23 | 0.87 | ACROS |
| α-Phellandrene | $C_{10}H_{16}$ | 136.23 | 0.65 | TCI |
| p-Cymene | $C_{10}H_{14}$ | 134.22 | 0.95 | TCI |
| Limonene | $C_{10}H_{16}$ | 136.25 | 0.97 | Sigma-Aldrich |
| Ocimene | $C_{10}H_{16}$ | 136.23 | 0.9 | Sigma-Aldrich |
| γ-terpinene | $C_{10}H_{16}$ | 136.23 | 0.95 | ACROS |
| Terpinolene | $C_{10}H_{16}$ | 136.26 | 0.85 | TCI |
| Linalool | $C_{10}H_{18}O$ | 154.25 | 0.97 | Sigma-Aldrich |
| Citral | $C_{10}H_{16}O$ | 152.23 | 0.9 | Sigma-Aldrich |
| Nopinone | $C_9H_{14}O$ | 138.1 | 0.98 | Sigma-Aldrich |
| β-Caryophyllene | $C_{15}H_{24}$ | 204.5 | 0.8 | Sigma-Aldrich |

**Table 9 : Chromatographic details for BPX-5 and DB-624 columns**

| DB-624 | | | | | | | | BPX-5 | | | | |
|---|---|---|---|---|---|---|---|---|---|---|---|---|
| ΔF (mL/min)/min | Flow (mL/min) | Hold (min) | Run time (min) | ΔT (°C/min) | Temperature (°C) | Hold (min) | Run time (min) | Pressure (PSI) | ΔT (°C/min) | Temperature (°C) | Hold (min) | Run time (min) |
| 0 | 4 | 1 | 1 | 0 | 40 | 8 | 8 | | 0 | 40 | 8 | 8 |
| 0.15 | 2 | 0 | 14.33 | 6 | 135 | 0 | 23.83 | | 6 | 135 | 0 | 23.83 |
| 0.15 | 3 | 10 | 31 | 1.25 | 180 | 0 | 59.83 | 24.3 | 0.6 | 145 | 0 | 40.50 |
| 0.2 | 5 | 1 | 70.67 | 6 | 250 | 3.5 | 75 | | 0 | 250 | 9 | 67 |

**Table 10 : Canister reproducibility (n=7)**

| Compounds | Concentration (ppb) | RSD (%) |
|---|---|---|
| Isoprene | 1880.4 | 5.1% |
| Toluene | 1244.9 | 3.1% |
| α-pinene | 815.3 | 4.1% |
| Camphene | 842.4 | 4.8% |
| Sabinene | 626.2 | 4.5% |
| Myrcene | 567.7 | 5.2% |
| β-pinene | 828.6 | 3.5% |
| 2-Carene | 812.8 | 4.9% |
| α-Phellandrene | 533.0 | 5.8% |
| 3-Carene | 745.1 | 4.2% |
| α-Terpinene | 713.4 | 3.5% |
| p-Cymene | 830.0 | 6.6% |
| Limonene | 820.4 | 4.9% |
| Ocimene | 717.4 | 9.8% |
| γ-Terpinene | 784.7 | 4.1% |
| Terpinolene | 729.9 | 5.2% |
| Linalool | 642.3 | 11.0% |
| Citral | 689.3 | 9.3% |
| Nopinone | 874.6 | 9.3% |
| β-Caryophyllene | 452.7 | 21.6% |

**Table 11 : Concentrations, relative standard deviations (RSD), memory effect, detection limits (DL) and concentrations of maximal relative residual, $\partial_{max}$ (in %), measurement at 80%RH (22°C) for the targeted compounds**

| Compounds | Concentration (ppb) | RSD (%) Laboratory | RSD (%) Field | Memory effect (%) | DL (ppt) | conc. $\partial_{max}$ (ppb) | $\partial_{max}$ |
|---|---|---|---|---|---|---|---|
| Isoprene | 4.5 | 2.2% | 2.2% | 1.5% | 12 | 1.3 | 44.2 |
| Toluene | 5.9 | 2.1% | | 0.5% | 10 | 3.5 | 5.7 |
| α-pinene | 4.0 | 2.0% | 1.4% | 4.3% | 4 | 1.2 | 12.0 |
| Camphene | 5.0 | 2.2% | 0.8% | 0.3% | 5 | 1.5 | 11.1 |
| Sabinene | 3.2 | 1.7% | 3.0% | 2.9% | 19 | 1.0 | 15.0 |
| Myrcene | 2.8 | 2.1% | 0.5% | 0.7% | 6 | 0.8 | 13.0 |
| β-pinene | 3.8 | 1.9% | 5.8% | 0.5% | 6 | 1.2 | 18.5 |
| 2-Carene | 3.8 | 2.3% | 1.1% | 0.6% | 5 | 1.1 | 9.7 |
| α-Phellandrene | 2.5 | 2.0% | 3.1% | 0.4% | 6 | 0.8 | 3.4 |
| $\Delta^3$-Carene | 3.5 | 2.0% | 3.5% | 0.2% | 5 | 1.0 | 11.3 |
| α-Terpinene | 3.5 | 2.4% | 2.2% | 0.2% | 8 | 1.0 | 30.0 |
| p-Cymene | 3.8 | 2.0% | 3.7% | 0.8% | 14 | 1.1 | 29.3 |
| Limonene | 3.7 | 2.2% | 1.3% | 0.3% | 4 | 1.1 | 14.5 |
| Ocimene | 3.3 | 2.8% | 2.7% | 0.2% | 4 | 1.0 | 26.9 |
| γ-Terpinene | 3.6 | 2.1% | 2.7% | 0.3% | 8 | 1.1 | 15.9 |
| Terpinolene | 3.5 | 2.1% | 2.9% | 1.5% | 9 | 1.0 | 16.8 |
| Linalool | 3.6 | 1.2% | 3.7% | 2.1% | 11 | 1.0 | 23.9 |
| Citral | 3.0 | 1.1% | 4.0% | 0.4% | 8 | 0.9 | 26.3 |
| Eucalyptol | 2.0 | - | 0.5% | - | 10 | - | - |
| Menthol | 4.1 | 1.5% | - | 2.4% | 9 | 1.0 | 95.6 |
| Nopinone | 4.7 | 2.3% | 3.9% | 2.9% | 7 | 1.5 | 27.3 |
| β-Caryophyllene | 2.1 | 1.1% | 6.9% | 0.6% | 9 | 3.2 | 14.6 |

**Table 12 : Statistics and uncertainties calculated at the mean value for BVOCs measured during the LANDEX field campaign in July 2017. The calibration technique used depending on the compounds is defined as (A) for the certified NPL gas standard and (B) for the generated mixture**

| Compounds | Min (ppt) | Max (ppt) | Moyen (ppt) | Incertitudes (ppt) | Calibration system used |
|---|---|---|---|---|---|
| β-pinene | 89 | 9902 | 1153 | 153 | A |
| α-pinene | 157 | 8928 | 1138 | 52 | A |
| Isoprene | <DL | 3616 | 408 | 26 | A |
| Myrcene | 20 | 1006 | 147 | 13 | A |
| 3-Carene | 16 | 1191 | 141 | 14 | A |
| Limonene | 9 | 1107 | 138 | 16 | A |
| p-Cymene | <DL | 842 | 124 | 35 | A (±20%) |
| Camphene | 17 | 924 | 120 | 79 | B |
| Linalool | <DL | 554 | 93 | 74 | B |
| Citral | 9 | 660 | 82 | 70 | B |
| Nopinone | <DL | 750 | 62 | 43 | B |
| Eucalyptol | <DL | 255 | 51 | 12 | A |
| Sabinene | <DL | 183 | 42 | 34 | B |
| β-Caryophyllene | <DL | 294 | 36 | 43 | B |
| Ocimene | <DL | 225 | 26 | 8 | A (±20%) |
| Terpinolene | <DL | 126 | 22 | 18 | B |
| α-Terpinene | <DL | 26 | 12 | 14 | B |
| γ-Terpinene | <DL | 41 | 10 | 12 | B |
| α-Phellandrene | <DL | 42 | 9 | 10 | B |
| 2-Carene | <DL | 8 | <DL | 8 | B |

**Figure 10:** Separation of 20 BVOCs from Table 2 using the DB-624 column (top) and BPX5 column (bottom). VOC mixing ratios were approximately 1 ppb.

**Figure 11:** Separation of 20 BVOCs together with 20 anthropogenic compounds. VOC mixing ratios were approximately 4 ppb. Compounds written in red character are BVOCs targeted

**Figure 12:** Investigation of the optimal desorption temperature for BVOC measurements. 2 replicates performed at each temperature (a) peaks area for the first desorption analysis and (b) desorption efficiency

**Figure 13:** Investigation of the safe sampling volume at 80% RH for the four most volatile compounds in the mixture

**Figure 14:** Relative deviation to BVOC measurements performed (i) with scrubber and without $O_3$, (ii) when $O_3$ (50 ppb) is added in the standard mixture without scrubber and (iii) when a scrubber is used with $O_3$. (a): KI scrubber and (b): thiosulfate scrubber. VOCs are classified from the less reactive (left) to the most reactive (right) with ozone. Errors bars correspond to 3 standard deviation values.

**Figure 15:** Canisters respond coefficient relative deviation to the NPL respond coefficient. Legend: canister creation date ddmmyyyy; canister used B: S053 and O: S052; n= number of replicates. Error bars = $1\sigma$

**Figure 16:** Uncertainties repartition for the five terms considered (precision, calibration, integration, memory effect and scrubber) at (a) 2 ppb and (b) 100 ppt with laboratory analytical parameters.

**Figure 17:** Uncertainty (k=2) repartition for the 6 terms considered (precision, calibration, integration, memory effect, scrubber and linearity) (upper panel) and concentration of $\beta$-pinene during the Landex-episode 1 field campaign (lower panel).

**Figure 18:** Times series of concentrations with their associated uncertainties (k=2) for a selection of BVOCs observed during the LANDEX campaign ($\beta$-pinene, $\alpha$-pinene, limonene, myrcene, $\Delta^3$-carene, linalool, isoprene and $\beta$-caryophyllene).