# Peer review of "Optimization of a gas chromatographic unit for measuring BVOCs in ambient air"

_Atmospheric Measurement Techniques, 2019_

## Referee Comment (RC1) · Anonymous Referee #2 · 8 Aug 2019

This paper described detail about biogenic VOC (terpenoid) measurements: method, calibration, error estimation etc. Accurate measurement of terpenoid is quite difficult, and most researchers avid detailed experiment, especially error estimation. Therefore, this paper would be quite informative and useful for terpenoid measurement, and for considering the limitation (error) of the observed terpenoid data.

Comments: Is it possible if other VOC (especially terpenoid) which is not included as target in this study would co-eluted with the target VOCs ? In the text (page11 line5), author explained about co-elution of target biogenic VOCs and anthropogenic VOCs. But the forest site like LANDEX campaign as demonstrated in this paper, non-target biogenic VOC would be more probable to interference the chromatogram. Are there any comment?

[Figure]

Small thing: page1 line23 & page2 line7 : "terpenoïds" -> "terpenoid"

---

## Referee Comment (RC2) · Anonymous Referee #3 · 13 Aug 2019

General Comments:

The paper describes an on-line gas chromatographic method which has been developed for the measurement of biogenic volatile organic compounds (BVOCs) in the atmosphere. It further offers a discussion of the issues relating to the measurement of BVOCs and focusses on the effect of ozone on the samples, evaluating three different options for ozone removal. The paper gives a nice comparison of the three selected ozone removal techniques and gives justification for selecting one of these for their system. Removal of ozone using heated stainless steel lines has not been considered here, which is a little frustrating since this appears to be the simplest ozone removal method to deploy, a short summary or comment regarding this method should be included. Co-elution of the targeted species with others commonly found in the

atmosphere is considered here, but I feel expanding this discussion would benefit the paper.

Specific comments:

Page 1, Line 19: Within the Abstract the authors state ". . . detection limit ranging from 4 ppt for $\alpha$-pinene to 19 ppt for sabinene." From the example chromatograms given in Figure1, they appear to have similar peak widths and both have ten carbon atoms and so in theory should elicit the same response from the FID detector. Could this indicate losses on surfaces during transfer of sabinene? Some discussion of why this range is so large would be interesting (perhaps under the "3.2 Thermodesorption" section.)

Page 1, Line 23: ". . .representing on average 60% of the measured terpenoïds." The authors should clarify whether this is 60% by mass or concentration.

Page 1, Line 23: "Uncertainties may be larger for the other compounds especially for those presenting a mixing ratio close to the detection limit." Isn't this the case for all compounds, including the BVOCs? As the detector signal approaches the limit of detection, the uncertainties increase? Some re-phrasing or clarification is needed here.

Page 2, Lines 14 - 18: This is true, but I'd also recommend including a statement that large uncertainties in observations, due to poor measurements of some species could also account for some of this gap, the effective removal of ozone in samples (as investigated by the authors) could help to address this.

Page 3, Line 12: Heated stainless steel lines have also been used to scrub ozone from samples (see Hakola et al. Atmos. Environ. (2012)) and arguably appears to be the simplest method to deploy. Did the authors consider using/testing this method? Most (if not all) GC systems contain heated stainless steel components so using this method and material should be suitable for observations of VOCs. If the authors have dismissed this method for any reason, it should be stated here since, on the face of it,

this is a "big miss" within the paper. Otherwise a statement regarding the potential use of this method should be included here.

Page 3, Lines 21 - 26: Two methods for quantifying terpenes are described, could the authors include here which method they chose and describe why?

Page 5, Lines 7 - 11: When discussing the relative humidity of the gases generated, the temperature of the lab should be stated since the absolute amount of water contained in a volume of gas will be dependent on temperature. For example, at 50 % RH, there is approximately twice as much water (by mass) in a volume of gas at 35°C as there is at 25°C.

3.1 Chromatographic separation, Pages 10 - 11: The authors have investigated co-elution of the targeted species with selected VOCs commonly found in Urban environments. It is difficult, or perhaps impossible, however to rule-out co-elution with species not contained in the 20 components the authors investigated here. A statement to this effect should be included here.

Specific typographic changes etc..:

Page 1, Line 16:

"Eluent was analysed using a flame ionization detection (FID)" Change to: "Eluent was analysed using flame ionization detection (FID)"

Page 1, Line 23: "...terpenoïds." Change to: "...terpenoids."

Page 1, Line 23:

"Uncertainties may be larger for the other compounds especially for those presenting a mixing ratio close to the detection limit."

Page 2, Line 23: "If this type of instrument..." Change to: "This type of instrument..."

Page 2, Line 24: "...the feasibility for ambient ..." Change to: "...but, the feasibility for

ambient . . ."

Page 12, Line 7: ". . .consistently with observations made . . ." Change to: ". . .consistent with observations made . . ."
* * *

---

## Author Comment (AC1) · 4 Oct 2019

Interactive comments – Report 1 (comments)

This paper described detail about biogenic VOC (terpenoid) measurements: method, calibration, error estimation etc. Accurate measurement of terpenoid is quite difficult, and most researchers avid detailed experiment, especially error estimation. Therefore, this paper would be quite informative and useful for terpenoid measurement, and for considering the limitation (error) of the observed terpenoid data.

Comments:

Is it possible if other VOC (especially terpenoid) which is not included as target in this study would co-eluted with the target VOCs? In the text (page11 line5), author ex-

plained about co-elution of target biogenic VOCs and anthropogenic VOCs. But the forest site like LANDEX campaign as demonstrated in this paper, non-target biogenic VOC would be more probable to interference the chromatogram. Are there any comment?

Reply This sentence has been added: "It should be noted that other compounds which have not been targeted here could possibly co-elute with targeted compounds and maybe other monoterpenes." Also, in the uncertainties calculation, the $u_\chi\_int^2$ term takes into account the possible evolution of peak shapes and possible over estimation due to the presence of a less abundant compound next to the eluted target compound.

Small thing:

page1 line23 & page2 line7 : "terpenoïds" -> "terpenoid" change have been made

---

## Author Comment (AC2) · 4 Oct 2019

Interactive comments – Report 2 (comments)

General Comments:

The paper describes an on-line gas chromatographic method which has been developed for the measurement of biogenic volatile organic compounds (BVOCs) in the atmosphere. It further offers a discussion of the issues relating to the measurement of BVOCs and focusses on the effect of ozone on the samples, evaluating three different options for ozone removal. The paper gives a nice comparison of the three selected ozone removal techniques and gives justification for selecting one of these for their system. Removal of ozone using heated stainless steel lines has not been

considered here, which is a little frustrating since this appears to be the simplest ozone removal method to deploy, a short summary or comment regarding this method should be included. Co-elution of the targeted species with others commonly found in the atmosphere is considered here, but I feel expanding this discussion would benefit the paper.

Specific Comments:

Page 1, Line 19: Within the Abstract the authors state "...detection limit ranging from 4 ppt for $\alpha$-pinene to 19 ppt for sabinene." From the example chromatograms given inFigure1, they appear to have similar peak widths and both have ten carbon atoms and so in theory should elicit the same response from the FID detector. Could this indicate losses on surfaces during transfer of sabinene? Some discussion of why this range is so large would be interesting (perhaps under the "3.2 Thermodesorption" section.)

Reply: The estimation of detection limit values mainly depends on the response coefficient of the considered compounds. The response coefficient of sabinene is different from other monoterpenes or isomers with which it had to match theoretically. Indeed, sabinene could be lost in the sampling line but it is not here the main reason for such difference. The most probable reason is the potential degradation of the sabinene in p-cymene and/or limonene during the thermodesorption, as demonstrated for Tenax and Carboxen by Coeur et al. (1997). Also, sabinene response coefficient values presented a high variability which is considered in the calibration term of the uncertainty calculation.

Change: Page 1, Line 23: "...representing on average 60% of the measured terpenoïds." Theauthors should clarify whether this is 60% by mass or concentration

Rely:Changed to "...representing on average 60% of the measured terpenoid concentration"

Comment: Page 1, Line 23: "Uncertainties may be larger for the other compounds

especially for those presenting a mixing ratio close to the detection limit." Isn't this the case for all compounds, including the BVOCs? As the detector signal approaches the limit of detection, the uncertainties increase? Some re-phrasing or clarification is needed here

Reply: OK. This sentence have been removed from the abstract

Comment: Page 2, Lines 14 - 18: This is true, but I'd also recommend including a statement that large uncertainties in observations, due to poor measurements of some species could also account for some of this gap, the effective removal of ozone in samples (as investigated by the authors) could help to address this.

Reply: the following sentence has been added : "Also, in those studies, a potential underestimation of concentration, high uncertainties on BVOC concentrations, and ozone reactivity (when no scrubber was used) could explain this missing reactivity"

Comment: Page 3, Line 12: Heated stainless steel lines have also been used to scrub ozone from samples (see Hakola et al. Atmos. Environ. (2012)) and arguably appears to be the simplest method to deploy. Did the authors consider using/testing this method? Most (if not all) GC systems contain heated stainless steel components so using this method and material should be suitable for observations of VOCs. If the authors have dismissed this method for any reason, it should be stated here since, on the face of it, this is a "big miss" within the paper. Otherwise a statement regarding the potential use of this method should be included here.

Reply: A comparison of our results with a KI scrubber compared to those obtained by Hellen et al. (2012) with a heated stainless steel tube of 3 m length (SS 3 m), at a flow rate of 1 L/min and with or without ozone, is reported in the table below. Without ozone, the recovery results with both types of scrubbers are comparable for toluene, nopinone, and monoterpenes except for terpinolene and camphene. $\beta$-caryophyllene and terpinolene recovery results are slightly better with the SS 3 m (103% and 104% respectively) than using KI scrubber (98% and 95% respectively). Linalool and camphene recovery results are slightly better with KI scrubber (93% and 96% respectively) than with the SS 3 m (87% and 91% respectively). With ozone, the monoterpenes, the $\beta$-caryophyllene and the nopinone recoveries are comparable or slightly better with SS 3 m than with KI scrubber. Linalool presents a bad recovery with the SS 3 m of 54% compared to 89% with KI scrubber. Here, we compare our results to the results of a SS 3 m but we probably used a SS 5 m length. The recoveries of $\beta$-pinene, linalool and $\beta$-caryophyllene with a SS 5 m and no ozone are smaller than with a SS 3 m with no ozone. As stated by Hellen et al. (2012), compound isomerization might be the reason for this. $\beta$-pinene is known to isomerize easily in myrcene and limonene during heating. For all those reasons, we preferred the use of a KI scrubber in our study.

Table : Average recoveries (%) of BVOCs for a KI scrubber and a heated stainless steel tube (SS) of 3 m length with and without 50 ppb of ozone, and a SS 5m length without ozone.

Change: P3L12 have been changed to. "... such as heated stainless steel tubes (Hellen et al., 2012), copper tubes ..."

P16L19 "In order to propose an exhaustive overview of ozone scrubber choice for BVOC measurements, a comparison of our results with a KI scrubber compared to those obtained by Hellen et al. (2012) with a heated stainless steel tube of 3 m length (SS 3 m), at a flow rate of 1 L/min and with or without 50 ppb of ozone, have been realised. Without ozone, the recovery results with both types of scrubbers are comparable for toluene, nopinone, and monoterpenes (94-97%), except for terpinolene and camphene. $\beta$-caryophyllene and terpinolene recoveries are slightly better with the SS 3 m (103% and 104% respectively) than with the KI scrubber (98% and 95% respectively). Linalool and camphene recoveries are slightly better using the KI scrubber (93% and 96% respectively) than with the SS 3 m (87% and 91% respectively). With ozone, the monoterpenes, $\beta$-caryophyllene and nopinone recoveries are comparable or slightly better with SS 3 m than with the KI scrubber (97-110%). Linalool presents a bad recovery with the SS 3 m of 54% compared to 89% with the KI scrubber. Here,

we compared our results to the results of a SS 3 m but we used a longest tube during the campaign, more like the SS 5 m length presented by Hellen et al. (2012). The recoveries of $\beta$-pinene, linalool and $\beta$-caryophyllene with a SS 5 m and no ozone are smaller than with a SS 3 m with no ozone. As stated by Hellen et al. (2012), the compound isomerization might be the reason for this. $\beta$-pinene is known to isomerize easily in myrcene and limonene during the heating step. For all those reasons, we finally preferred the use of a KI scrubber in our study." have been added.

Comment: Page 3, Lines 21 - 26: Two methods for quantifying terpenes are described, could the authors include here which method they chose and describe why?

Reply: In section 2.1, we answered this question in describing our standard system.

Comment: Page 5, Lines 7 - 11: When discussing the relative humidity of the gases generated, the temperature of the lab should be stated since the absolute amount of water contained in a volume of gas will be dependent on temperature. For example, at 50 % RH, there is approximately twice as much water (by mass) in a volume of gas at 35âŮęC as there is at 25âŮęC.

Reply: Room temperature for those experiments was 22°C.

Comment: 3.1 Chromatographic separation, Pages 10 - 11: The authors have investigated co-elution of the targeted species with selected VOCs commonly found in Urban environments. It is difficult, or perhaps impossible, however to rule-out co-elution with species not contained in the 20 components the authors investigated here. A statement to this effect should be included here.

Reply: The following sentence has been added: "It should be noted that other compounds which have not been targeted here could co-elute with targeted compounds and maybe other monoterpenes." Also, in the uncertainty calculation, the $u\chi\_int\hat{\ }2$ term takes into account the possible evolution of peak shapes and the possible overestimation due to the presence of a less abundant compound next to the eluted target

compound.

Specific typographic changes etc..:

Page 1, Line 16:"Eluent was analysed using a flame ionization detection (FID)" Changed to: "Eluent was analysed using flame ionization detection (FID)"

Page 1, Line 23: "...terpenoïds." Changed to: "...terpenoids."

Page 1, Line 23:"Uncertainties may be larger for the other compounds especially for those presenting a mixing ratio close to the detection limit." Sentence deleted

Page 2, Line 23: "If this type of instrument..." Changed to: "This type of instrument..."

Page 2, Line 24: "...the feasibility for ambient..." Changed to: "...but, the feasibility for ambient..."

Page 12, Line 7: "...consistently with observations made..." Changed to: "...consistent with observations made..."

Reference:

Coeur, C., Jacob, V., Denis, I., Foster, P. : Decomposition of $\alpha$-pinene and sabinene on solid sorbents, tenax TA and carboxen. J. Chromatogr. A 786, 185–187. https://doi.org/10.1016/S0021-9673(97)00562-1, 1997.

Hellén, H., Kuronen, P., Hakola, H. : Heated stainless steel tube for ozone removal in the ambient air measurements of mono- and sesquiterpenes. Atmos. Environ. 57, 35–40. https://doi.org/10.1016/j.atmosenv.2012.04.019, 2012.

| | with ozone | | No ozone | | |
|---|---|---|---|---|---|
| | KI scrubber | SS (3 m) | KI scrubber | SS (3 m) | SS (5 m) |
| Toluene | 98 | 108 | 96 | 95 | 100 |
| α-pinene | 98 | 102 | 97 | 94 | 101 |
| Camphene | 98 | 111 | 96 | 91 | 108 |
| β-pinene | 97 | 97 | 96 | 94 | 90 |
| 3-Carene | 97 | 107 | 96 | 96 | 103 |
| p-Cymene | 102 | 102 | 97 | 95 | 107 |
| Limonene | 99 | 110 | 97 | 96 | 121 |
| Terpinolene | 87 | 101 | 95 | 104 | 102 |
| Linalool | 89 | 54 | 93 | 87 | 8 |
| Nopinone | 101 | 103 | 97 | 95 | 100 |
| β-Caryophyllene | 60 | 68 | 98 | 103 | 66 |

**Fig. 1.**